# Enhanced secondary organic aerosol formation from the photo-oxidation of mixed anthropogenic volatile organic compounds

Junling Li[1], Hong Li[*,1], Kun Li[2], Yan Chen[3], Hao Zhang[1], Xin Zhang[1], Zhenhai Wu[1], Yongchun Liu[4], Xuezhong Wang[1], Weigang Wang[3], Maofa Ge[3]

[1].State Key Laboratory of Environmental Criteria and Risk Assessment, Chinese Research Academy of Environmental Sciences, Beijing 100012, China

[2] Laboratory of Atmospheric Chemistry, Paul Scherrer Institute, 5232 Villigen, Switzerland

[3]State Key Laboratory for Structural Chemistry of Unstable and Stable Species, Beijing National Laboratory for Molecular Sciences, CAS Research/Education Center for Excellence in Molecular Sciences, Institute of Chemistry, Chinese Academy of Sciences, Beijing 100190, China

[4] Beijing Advanced Innovation Center for Soft Matter Science and Engineering, Beijing University of Chemical Technology, Beijing 100029, China

*Correspondence to*: Hong Li (lihong@craes.org.cn)

**Abstract.** Vehicular exhaust is one of the important contribution sources of secondary organic aerosol (SOA) in urban areas. Long-chain alkanes and aromatic hydrocarbons are included in gaseous organic pollutants of vehicle emissions, the representative for diesel and gasoline vehicles respectively. In this work, the SOA production from individual anthropogenic volatile organic compounds (AVOCs) (n-dodecane, 1,3,5-trimethylbenzene) and mixed AVOCs (n-dodecane + 1,3,5-trimethylbenzene) were studied with a large-scale outdoor smog chamber. Results showed that the SOA formation from the mixed AVOCs was enhanced compared to the predicted SOA mass concentration based on the SOA yield of individual AVOCs. According to the results of mass spectrometry analysis with electrospray ionization time-of-flight mass spectrometry (ESI-ToF-MS), interaction occurred between intermediate products from the two precursors, which could be the main reason for the enhanced SOA production from the mixed AVOCs reaction system. The study results could improve our understanding about the contribution of representative precursors from the vehicular exhaust to the formation of SOA in urban areas. This study also indicates that further studies on SOA chemistry from the mixed VOCs reaction system are needed, as the interactions between them and the effect on SOA formation can give us a further understanding of the SOA formed in the atmosphere.

## 1 Introduction

Secondary organic aerosol (SOA) has received considerable attention during the past few decades, as it plays an important role in affecting global climate change (Shrivastava et al., 2017; von Schneidemesser et al., 2015; Mellouki et al., 2015; Kanakidou et al., 2005), atmospheric visibility (Zhang et al., 2015a; Moise et al., 2015; Laskin et al., 2015; Ren et al., 2018), and public health (Poschl, 2005; Poschl and Shiraiwa, 2015; Zhang et al., 2016; Requia et al., 2018). The formation, growth, and transformation of SOA influence the atmospheric aerosol's physicochemical properties (Poschl and Shiraiwa, 2015;

Moise et al., 2015; Mellouki et al., 2015; Herrmann et al., 2015). The precursors of SOA mainly include anthropogenic

volatile organic compounds (AVOCs) and biogenic volatile organic compounds (BVOCs) (Kelly et al., 2018); in urban areas, AVOCs are the main sources of SOA, e.g., gasoline vehicle emissions (Johnson et al., 2004; Charron et al., 2019; Yang et al., 2018), diesel vehicle emissions (Paulsen et al., 2005; Wirtz and Martin-Reviejo, 2003; Odum et al., 1996; Zhao et al., 2015), and solvent use (Li et al., 2017c; Kansal, 2009).

    Early regional air quality models underestimated the observed SOA concentrations in large areas of the atmosphere

(Volkamer et al., 2006; Heald et al., 2005; de Gouw et al., 2005; Appel et al., 2017; Huang et al., 2017a); after incorporating the newly discovered SOA sources, the gap between the observed and predicted SOA concentrations is decreasing (Zhao et al., 2016; Slowik et al., 2010; Hodzic et al., 2010). The SOA formation processes in the atmosphere are very complicated; although the degradation of most VOCs is clear now, the formation and aging of a large amount of SOA is still unclear. Previous studies found that the observed organic aerosol concentration could not be explained by the traditional yields of the

measured AVOCs (de Gouw et al., 2005); in addition, field observations found that the potential interactions between AVOCs and BVOCs existed during SOA formation (Spracklen et al., 2011; Hoyle et al., 2011; Glasius et al., 2011; Galloway et al., 2011; Kari et al., 2019): AVOCs could enhance (Spracklen et al., 2011; Carlton et al., 2010; Shilling et al., 2013) or suppress SOA formation from BVOCs (Kari et al., 2019). A recent study also found that the SOA formation could be reduced by the mixture of BVOCs (McFiggans et al., 2019). These findings indicate that there are interactions in the

complex mixtures of VOCs, which may influence the SOA production estimation if they were considered in models.

    In urban areas, the vehicular exhaust is one of the important sources of SOA, the representative substances of which include aromatic hydrocarbons and long-chain alkanes (Paulsen et al., 2005; Wirtz and Martin-Reviejo, 2003; Charron et al., 2019; Saathoff et al., 2009; Zhao et al., 2015; Gentner et al., 2012). As an important contributor to SOA in urban areas, aromatic hydrocarbons are generally concerned about their kinetics (Atkinson and Arey, 2003; Calvert et al., 2002), reaction

mechanisms (Tsiligiannis et al., 2019; J. Midey∗ et al., 2003; Huang et al., 2017c; Wang et al., 2020; Garmash et al., 2019), SOA yield (Cao and Jang, 2007; Kroll et al., 2007; Ng et al., 2007b; Huang et al., 2017b), ozone generation potential (Luo et al., 2019), and SOA physicochemical properties (optical properties, morphology, etc.) (Grosjean, 1981; Li et al., 2018; Li et al., 2017b; Phillips and Smith, 2014; Kim and Paulson, 2013; Huang et al., 2018). Long-chain alkanes, as representative substances of intermediate volatile organic compounds (IVOCs), are considered as a potential contributor of SOA (Robinson

et al., 2007; Trostl et al., 2016; Shiraiwa et al., 2013). The studies about long-chain alkanes include SOA chemical compositions (Fahnestock et al., 2015; Yee et al., 2013; Aimanant and Ziemann, 2013; Yee et al., 2012), SOA yield (Loza et al., 2014; Tkacik et al., 2012), and SOA optical properties (Li et al., 2017a; Li et al., 2020), etc. The aromatic hydrocarbons and long-chain alkanes are generally studied separately in the laboratory. However, it should be noted that in the real atmosphere, aromatic hydrocarbons and long-chain alkanes often exist at the same time, especially from vehicle emissions

(Wu and Xie, 2018). The studies that cover these two types of substances in one reaction system are still limited, and the corresponding SOA formation and reaction processes are not yet clear.

In this work, 1,3,5-trimethylbenzene and n-dodecane are selected as representative substances. As the concentration of 1,3,5-TMB is much higher than that of n-dodecane in both the gasoline compositions and ambient air, the initial concentration ratio of 1,3,5-TMB and n-dodecane in this work is about 10:1 (ppbv): Schauer et al. (2002) reported that 1,3,5-TMB and n-dodecane in the gasoline composition were about 7450 and 136 μg/g, respectively; Gentner et al. (2012) reported that the weight percentage of 1,3,5-TMB and n-dodecane in liquid gasoline were 0.530-0.881 and 0.004-0.045 (% weight by carbon), respectively. According to field observations in China, the measured 1,3,5-TMB concentration at the rural site in the YelRD (Yellow River Delta) region in 2017 could reach 1.447 ppb (Chen et al., 2020), and the measured $C_{12}$ alkane concentration was $0.122 \pm 0.12$ ppb at PRD (Pearl River Delta) region, and $0.129 \pm 0.086$ ppb at NCP (North China Plain) region in 2018 (Wang et al., 2020).

The work aims to investigate the SOA formation from the mixed AVOCs reaction system. In this study, the SOA yield derived from n-dodecane and 1,3,5-trimethylbenzene in the presence of HONO were obtained with a large-scale outdoor smog chamber, and the SOA derived from the mixed AVOCs were measured. The measured SOA mass concentration from mixture AVOCs reaction system was compared to the predicted SOA mass based on the SOA yield of n-dodecane and 1,3,5-trimethylbenzene. SOA particles were collected and analyzed with an electrospray ionization time-of-flight mass spectrometer (ESI-ToF-MS) to achieve insight into the chemical composition and interactions. The results here are helpful to improve our understanding of the contribution of representative precursors from vehicle exhaust to SOA.

## 2 Experimental Section

### 2.1 Experimental Methods

The experiments were conducted in a 56 m³ (3.2 m × 6.2 m × 2.5 m) outdoor smog chamber constructed at Chinese Research Academy of Environmental Sciences (the CRAES Chamber, 40°02'27.73'N, 116°24'41.56'E). The details of the chamber had been described previously (Li et al., 2021). Briefly, the chamber was made of FEP Teflon film, the light transmission of which was above 90% at the wavelength of 350-900 nm. The substances inside the chamber could be mixed well within 4 min. The experimental duration under solar irradiation was about 7~8 h. After each experiment, the chamber was cleaned with zero air for at least 24 h with a flow rate of 200 L/min.

1,3,5-trimethylbenzene or n-dodecane was introduced into the chamber by zero air through the custom-made U-shaped glass tube with a known volume of liquid 1,3,5-trimethylbenzene or n-dodecane. Concentrations of 1,3,5-trimethylbenzene and n-dodecane were measured before and after reactions by collecting samples on Tenax TA solid adsorbent and analyzing by thermal desorption-gas chromatography with flame ionization detection (TD, UNITY-xr; GC, 7890B). The OH precursor of the experiments was HONO, it was prepared by dropwise addition of 1 mL 2 wt% $NaNO_2$ solution into 2 mL 15 wt% sulfuric acid solution in a custom-made glass bubbler, the bubbler was attached to the smog chamber with a Teflon tube, and the formed HONO was introduced into the chamber by zero air. The NO, $NO_2$, and formed ozone in the chamber were

measured by NOx analyzer (EC 9841, ECOTECH, Australia) and ozone analyzer (EC 9830, ECOTECH, Australia), respectively. After the gas species mixed evenly in the chamber, the enclosure of the chamber was opened.

After each photochemical experiment, the formed aerosol particles in the chamber were collected by a low flow sampler (LV 40BW, Sibata Scientific Technology Ltd., Soka, Japan) at a flow rate of 5 L/min for 10 min. The PTFE filters (0.2 μm, 47 mm, MerckMillipore, TYPE FGLP) used were extracted in 5 mL methanol sonicating for 30 min. The methanol solutions were analyzed by an ESI-TOF-MS (Bruker, Impact II) in positive mode, and the chemical compositions of the formed SOA were obtained. The methanol solutions were also detected with a UV−Vis light spectrometer (Hitachi, U-3900), which was

used to detect the absorbing property of the formed SOA. The Attenuated Total Internal Reflection Infrared (ATR-IR) analysis was applied to determine the potential functional groups in SOA extracts, an FTIR spectrometer (Bruker, Tensor 27) equipped with a RT-DLaTGs detector was used. The SOA extracts were deposited and dried directly on the Diamant crystal of an ATR-IR cell. The spectra of the dry SOA extracts were recorded by using a background spectrum obtained with no samples as the reference (100 scans, 2.4 cm$^{-1}$ resolution).

The chemicals following were used without further purification: 1,3,5-trimethylbenzene (1,3,5-TMB) (99%, Acros), n-dodecane (>99%, Alfa Aesar), sulfuric acid (>95%, Beijing Chemical Works), sodium nitrite (98%, Alfa Aesar), methanol (99.9%, Merck), acetonitrile (99.8%, Fisher Chemical).

## 2.2 Calculation Methods

### 2.2.1 Wall-Loss Corrections

As SOA yields could be underestimated due to the losses of SOA forming vapors to chamber walls, the vapor wall-loss was considered and corrected in this work (Zhang et al., 2014). The competition between the uptake of organic vapor by the chamber walls and the aerosol particles would determine the effect of vapor wall-loss on SOA yields (Zhang et al., 2015b). The ratio of average gas-particle partitioning timescale ($\bar{\tau}_{g-p}$) to the vapor wall-loss timescale ($\bar{\tau}_{g-w}$) could be used to evaluate the underestimation of SOA yields (Zhou et al., 2011; Chen et al., 2019).

The average gas-particle partitioning timescale ($\bar{\tau}_{g-p}$) could be expressed as the following equation (Seinfeld J.H., 2006; Zhang et al., 2014):

$$\bar{\tau}_{g-p} = \frac{1}{2\pi \bar{N}_p \bar{D}_p D_{gas} \bar{F}_{FS}} \tag{1}$$

where $\bar{N}_p$ was the average number concentration of the formed particles during the experiment, $\bar{D}_p$ was the number mean diameter of the particles, $D_{gas}$ was the gas-phase diffusivity, $\bar{F}_{FS}$ was the Fuchs-Sutugin correction for noncontinuum mass

transfer (Seinfeld J.H., 2006).

       The vapor wall-loss timescale ($\bar{\tau}_{g-w}$) could be expressed as the following equation (Zhang et al., 2015b):

$$\bar{\tau}_{g-w} = \frac{1}{k_w} \tag{2}$$

$$k_w = \left(\frac{A}{V}\right) \frac{a_w \frac{\bar{c}}{4}}{1.0 + \frac{\pi}{2}\left[\frac{a_w \bar{c}}{4(k_e D_{gas})^{0.5}}\right]} \tag{3}$$

where $k_w$ was the wall loss rates of the organic vapor; $\frac{A}{V}$ was the ratio of surface to volume of the chamber, 1.55 m$^{-1}$ for this chamber; $a_w$ was the mass accommodation coefficient of vapors deposition to the wall ($10^{-5}$ was used here) (Zhang et al., 2014); $\bar{c}$ was the root mean square speed of the gas; $k_e$ was the eddy diffusion coefficient, which was set to 0.12 s$^{-1}$ according to the reported values for a 60 m$^3$ chamber (McMurry and Grosjean, 1985). The detailed calculation of $\bar{c}$, $D_{gas}$, $k_n$ and $\bar{F}_{FS}$ were shown in the Supporting Information. The uncertainty of the mass correction here is about ±11.2% (see Supporting Information for details).

Particle wall-loss to chamber walls would also cause underestimation when calculating the SOA yield from measurements if these losses were not corrected for. Thus particle wall-loss was accounted for during the experiments. The particle growth data was corrected for wall-loss, in which size-dependent coefficients from inert particle wall-loss experiments (ammonium sulfate) were applied to the particle volume data (Li et al., 2021):

$$k_{dep}(d) = 6.35 \times 10^{-6} d^{1.56} + \frac{6.38}{d^{0.67}} \tag{4}$$

where $k_{dep}(d)$ was the wall-loss loss coefficient of particles in the diameter $d$.

### 2.2.2 SOA Yields

The secondary organic aerosol (SOA) yield (Y) was defined as the fraction of a reactive organic gas (ROG) that was converted to aerosols, and it could be calculated by the following equation:

$$Y = \frac{\Delta M_o}{\Delta ROG} \tag{5}$$

where $\Delta M_o$ (µg m$^{-3}$) was the mass concentration of the organic aerosol, and $\Delta ROG$ (µg m$^{-3}$) was the amount of the ROG reacted.

For the mixed anthropogenic volatile organic compounds (AVOCs), the formed SOA mass was predicted based on the SOA precursors and their SOA yield measured in this study. The possible non-linear interactions between the anthropogenic VOC mixtures were not taken into account. Specifically, the calculation equation (Kari et al., 2019) could be expressed as follows:

$$SOA_{predicted} = \sum_i (\Delta ROG_i \times Y_i) \tag{6}$$

where $\Delta ROG_i$ was the amount of the $ROG_i$ reacted, and $Y_i$ was the SOA yield of $ROG_i$.

### 3 Results and Discussion

A set of experiments are conducted in summer, of which the initial conditions and general results are shown in Table 1. The experiments are conducted as follows: n-dodecane + HONO; 1,3,5-trimethylbenzene + HONO; n-dodecane + 1,3,5-trimethylbenzene + HONO. The experiments are conducted under similar conditions, and the details of the relative humidity

(RH), temperature (T), and the $NO_2$ photolysis rate ($J(NO_2)$) of the experiments are shown in Figure S1. Ng et al. (2007a) reported that the efficient photolysis of HONO (the same method with this study) could generate relatively high concentrations of OH, 1 ppm NOx ~ $2 \times 10^7$ molecules/cm$^3$ OH initially. The $NO_X$ concentration applied in this work is in the range of 190-260 ppb, resulting in the estimated OH concentration being (4 - 5.2) $\times 10^6$ molecules/cm$^3$ in the pure and mixture experiments. The $NO_2$ photolysis rates of the experiments at noon in summer are in the range of 0.005-0.006 s$^{-1}$; for the experiment MIX-3, the weather is cloudy, and the $J(NO_2)$ at noon is relatively smaller, 0.004 s$^{-1}$. The temperature in summer at noon is in the range of 30-46 ℃, the RH inside the chamber is < 10%. The reaction profiles of photo-oxidation of n-dodecane, 1,3,5-TMB, and mixture AVOCs under HONO conditions in summer are shown in Figure 1. According to a previous study (Chen et al., 2019), the formed inorganic nitrate is negligible for the high-NOx oxidation of gasoline, in which the experimental conditions are similar to this study (NOx 130 ppb, formed aerosol mass concentration 34.6 μg/m$^3$). In the pure and mixture experiments, the $NO_X$ concentration is equivalent, so the formed nitric acid should be similar. Therefore, the increase in particle mass concentration in the mixture experiments is likely from the organic aerosols.

The SOA yields of n-dodecane and 1,3,5-TMB are 14.1~23.1% and 1.1~2.4%, respectively, as shown in Table 1. The predicated SOA mass derived from the mixture of these VOCs is based on the measured SOA yields of n-dodecane and 1,3,5-TMB, without considering possible non-linear interactions between them. Then the observed SOA mass is compared to the predicted values. It can be seen that nearly all the measured values are higher than the predicted SOA mass both before and after wall-loss correction. In other words, the SOA formation is enhanced when the two AVOCs are mixed, indicating the potential synergistic effect may exist in the mixture AVOCs reaction system. The findings above would be discussed further in the following parts.

**3.1 Enhancement of SOA formation**

Figure 1 shows the formation and evolution of the SOA during the photochemical reaction processes in summer. The number mean diameter, number concentration, surface mean diameter, total surface, and mass concentration of the particles are analyzed and compared. The number mean diameters of the formed particles from n-dodecane, 1,3,5-TMB, and the mixture are 100 nm, 50-100 nm, and 150-200 nm, respectively. This suggests that after mixing the two precursors, the number mean diameter of the formed particle became larger. The number concentration of the formed particles, similarly, increased from $2.0 \times 10^3$ #/cm$^3$ for single precursors to above $1.0 \times 10^4$ #/cm$^3$ for the mixture. Because of the enhanced particle number concentration and diameter, the mass concentration of particles increases from < 4 μg/m$^3$ for individual precursors to > 40 μg/m$^3$ for the mixture. It can be seen that the mass concentration of SOA generated by the mixture AVOCs system is significantly higher than the sum of the SOA generated by the two separate systems. It should be noted that the surface mean diameter of the particles from n-dodecane, 1,3,5-TMB, and the mixed AVOCs is all around 200 nm. However, due to the enhanced number concentration for mixture, the total surface of the formed particles for the mixture (>$1.0 \times 10^9$ nm$^2$/cm$^3$) was higher than individual precursors (<$1.0 \times 10^8$ nm$^2$/cm$^3$ for n-dodecane; <$5.0 \times 10^7$ nm$^2$/cm$^3$ for 1,3,5-TMB). Overall, after the two precursors are mixed, the number mean diameter, number concentration, total surface,

and mass concentration of the generated particles were improved, while the surface mean diameter of the particles did not change.

From the results shown above, we know that the SOA yield is significantly enhanced when mixing n-dodecane and 1,3,5-TMB. Experimental conditions can influence the SOA yields; however, the effect is not obvious. First, precursor concentration may play a role; however, we can rule it out based on the analysis below. Several previous studies (Lauraguais

et al., 2012; Loza et al., 2014; Zhou et al., 2011; Ng et al., 2007a) have reported that the aerosol formation is strongly affected by the initial precursor concentration, with a higher initial concentration of precursor leading to higher SOA yields. As higher initial precursor concentration will produce a higher amount of condensable products through chemical processes, thus the formed SOA mass will be higher. The aerosol present in the system will directly affect the gas-particle partitioning, as the medium, it can adsorb the oxidation products; thus, higher SOA mass will lead to higher SOA yield (Lauraguais et al.,

2014). In this work, for TMB-1 and TMB-2, while keeping the HONO concentration basically unchanged, the concentration of 1,3,5-TMB increases from 105 ppb (514.5 $\mu g/m^3$) to 178 ppb (882.9 $\mu g/m^3$), the yield increases by only 0.9%. For the mixture AVOCs reaction system, the concentrations of the precursors for MIX-1 and MIX-2 are 168 and 155 ppb (824.2 and 756.2 $\mu g/m^3$, 1,3,5-TMB), 28 and 22 ppb (194.7 and 152.3 $\mu g/m^3$, n-dodecane), respectively. Compared with TMB-2 experiments, the concentration of MIX precursor is only increased about 136 $\mu g/m^3$ and 25.6 $\mu g/m^3$ (3%-15%). According to

the SOA yields of TMB reaction system, the increase in the precursors mass concentration of the mixture system is not the reason for the large increase in the SOA mass concentration. Second, NOx may also influence the SOA yield, but likely not the case here. Tsiligiannis et al. (2019) observed that the particle formation strongly varied with $NO_X$ conditions, the increasing $NO_X/\Delta TMB$ ratio would suppress the SOA formation. In this work, regardless of whether it is a single or a mixture reaction system, the $NO_X$ concentration in the system remains basically unchanged. For experiments TMB-2, MIX-1,

and MIX-2, they have a similar $\Delta VOC/NO_X$ ratio, all around ~8, but the formed SOA mass concentration is quite different. This indicates that the $\Delta VOC/NO_X$ ratio here has little effect on the enhanced SOA mass concentration of the mixed AVOCs reaction system.

Rate constants for the reactions of n-dodecane and 1,3,5-TMB with OH radical at 298 K are 13.2 × 10^-12 $cm^3$ molecule$^{-1}$ s$^{-1}$ and 56.7 × 10^-12 $cm^3$ molecule$^{-1}$ s$^{-1}$, respectively (Atkinson and Arey, 2003). As shown in Table S3, OH

reactivity of Dod-1 and Dod-2 is about 6.5-7.1 s$^{-1}$; OH reactivity of TMB-2, and TMB-3 is in the range of 237.1-248.3 s$^{-1}$; and OH reactivity of MIX-1, MIX-2, MIX-3, and MIX-4 is in the range of 223.3~361.5 s$^{-1}$. This indicates that OH reactivity of the mixture experiments differs greatly from that of dodecane experiments, but it is very close to that of 1,3,5-TMB experiments. However, the mixture experiments still have a large enhancement in SOA formation compared with 1,3,5-TMB experiments, indicating that this enhancement is likely not due to the different OH reactivity. Rate constants for the reactions

of $NO_2$ and NO with OH radical at 298 K are 4.1× 10^-11 $cm^3$ molecule$^{-1}$ s$^{-1}$ and 3.3× 10^-11 $cm^3$ molecule$^{-1}$ s$^{-1}$, respectively (Atkinson et al., 2004). The OH reactivity of NOx is similar for all experiments (189.6-238.8 s$^{-1}$), and therefore likely plays a minor role in influencing SOA concentration.

For the enhancement in SOA yield of the mixture AVOCs system, we propose two possible conjectures, as revealed in Figure 2. The first conjecture is that the gas-particle partitioning of the system has changed. The SOA yield of n-dodecane (14.1-23.1%) is significantly higher than that of 1,3,5-TMB (1.1-2.4%), so the volatility of its products (including gas phase and particle phase) is relatively lower, and it is easier to form particles, e.g., nucleation; for the 1,3,5-TMB reaction system, the products have higher volatility and are difficult to condense and nucleate, so the yield is lower. When 1,3,5-TMB is mixed with n-dodecane, the products of n-dodecane provide a lot of particles for the products of 1,3,5-TMB to condense, so the yield is greatly improved. Another conjecture is that there are chemical interactions between the two systems, i.e., the intermediate products of the two precursors may react with each other.

In order to know which conjecture is correct, different injection experiments are performed: n-dodecane and HONO are introduced into the chamber firstly, and after one (Figure 3c and d, MIX-6) or four (Figure 3a and b, MIX-7) hours of photochemical reaction, 1,3,5-TMB is introduced into the chamber. As shown in Figure 3a and c, after the introduction of 1,3,5-TMB, the mass and number concentration of the particles has a certain increase, and the consumption of $NO_x$ is accelerated. However, compared with Figure 3e (n-dodecane and 1,3,5-TMB are added together before the photochemical experiments, MIX-4), the final SOA mass concentration of MIX-6 and MIX-7 (Figure 3a and c) are much lower. If our first conjecture plays an important role, one would expect large SOA mass enhancement (similar with mixed experiments) as the products of n-dodecane provide enough condensational sink for 1,3,5-TMB products to condense. The result here indicates that the gas-particle partitioning conjecture plays a minor role in the SOA yield enhancement. To further verify our second conjecture, the particle compositions are analyzed and shown below.

Figure 4 shows the ESI-ToF-MS mass spectra of SOA generated from n-dodecane, 1,3,5-TMB, and mixture AVOCs. The representative identified products with strong intensity are shown in Figure S2 and Table 2. The identified products are mainly based on the mass spectra and previous related studies (Tsiligiannis et al., 2019; Li et al., 2017a; Sato et al., 2019). As shown in Figure 4, most products from n-dodecane, 1,3,5-TMB, and mixture AVOCs SOA are concentrated around $m/z$ 200-450, in the range of $m/z$ 500-700, oligomers are formed.

Huang et al. (2015) reported that the predominant products for the aging of 1,3,5–trimethylbenzene secondary organic aerosol were organic nitrogen-containing products, aromatic organic acid, oxocarboxylic acid, and oligomer compounds. Due to the various $NO_x/\Delta TMB$ ratios, the formed products might be different. Usually, higher NO would lead to the suppression of oligomers and particle formation, and higher NO would increase the formation of organonitrates (Tsiligiannis et al., 2019). The products derived from n-dodecane in the presence of $NO_x$ were mainly oxygen-containing organic compounds (i.e., peroxyhemiacetals, hemiacetals, esters, aldol condensation) and organonitrate products (Fahnestock et al., 2015; Lim and Ziemann, 2005). As shown in Table 2, the products derived from 1,3,5-TMB are mainly organonitrates and oxygen-containing organic compounds. For products derived from n-dodecane, the main components are also oxygen-containing organic compounds and organic nitrates. It should be noted that in the mixture AVOCs system, some products that are not detected in the separate reaction system (n-dodecane or 1,3,5-TMB), such as $C_{16}H_{30}O_4$, $C_{16}H_{24}O_5$, $C_{29}H_{48}O_{10}$, $C_{35}H_{68}O_{10}$, etc. This indicates that interactions occur between the intermediate products from n-dodecane and 1,3,5-TMB.

The gas-phase products of OH-initiated oxidation of 1,3,5-TMB in the presence of $NO_x$ are mainly 3,5-dimethyl benzaldehyde ($C_9H_{10}O$), 3,5-dimethylbenzoic acid ($C_9H_{10}O_2$), 2-methyl-4-oxo-2-pentenal ($C_6H_8O_2$), 2-methyl-4-oxo-2-pentenoic acid ($C_6H_8O_3$), 2,4,6-trimethylphenol ($C_9H_{12}O$), and 3,5-dimethyl-2-furanone ($C_6H_8O_2$) (Huang et al., 2015), which contain carbonyl or hydroxyl groups that are formed within 1h photochemical reaction. The intermediate products of OH-initiated oxidation of n-dodecane in the presence of $NO_x$ are also compounds containing carbonyl and hydroxyl groups, and more alcohol can be formed due to $RO_2 + NO$ reaction compared to low $NO_X$ condition (Fahnestock et al., 2015). These compounds tend to undergo acetal reaction and/or esterification reaction in the particle phase. When the photochemical reaction is initiated, the intermediate products produced by 1,3,5-TMB and n-dodecane exist in the same reaction system, acetal and esterification reactions are more likely to occur in the particle phase due to higher concentration of aldehydes, ketones, alcohols, and carboxylic acids. The proposed reaction mechanism of the mixture AVOCs system is shown in Figure 5. As an example, the $C_{16}H_{24}O_5$, which has a much higher intensity in the mixed AVOCs system (as shown in Table 2 and discussed above), might be an ester from the reaction of an acid and an alcohol from 1,3,5-TMB and n-dodecane, respectively.

## 3.2 Light absorption of secondary organic aerosol

Figure 6 (a) shows the UV-Vis spectra of the n-dodecane, 1,3,5-TMB, and mixture AVOCs SOA filter extract. Before analyzing the samples, the blank PTFE membrane filter is dissolved with methanol, and the ultraviolet-visible absorption spectrum of the methanol solution is analyzed. The absorption of the SOA filter solutions is mainly concentrated in the wavelength of < 300 nm. This is consistent with previous literature reports: Li et al. (2017a) found that the n-dodecane SOA solutions had no detectable absorption in the wavelength of > 350 nm;Huang et al. (2018) found that the 1,3,5-TMB SOA solutions also had no obvious absorption in the wavelength of > 300 nm. Based on the light absorption spectra, the mass absorption efficiency (MAE, $m^2$/g) of the SOA in the extracts is calculated using the following equation (Chen et al., 2016):

$$MAE(\lambda) = ln(10) \, Abs \, (\lambda)/C_{OM} \qquad (7)$$

where $Abs(\lambda)$ is the light absorption coefficient ($m^{-1}$), and $C_{OM}$ is the SOA mass concentrations in the extracts. The MAE of the SOA extracts in Figure 6 (a) was calculated from 200 to 300 nm. The MAE at 205 nm were in the order: 1,3,5-TMB SOA (56.8 $m^2$/g) > dodecane SOA (42.5 $m^2$/g) > mixture AVOCs SOA (19.5 $m^2$/g). The MAE in the 210-250 nm band also shows the same pattern. This indicates that the SOA generated by the mixture AVOCs contains less light-absorbing substance per unit mass relative to dodecane SOA and 1,3,5-TMB SOA.

To further determine the potential functional groups in SOA extracts, ATR-IR spectra were acquired (Figure 6b). To eliminate the influence of water, experiments were conducted under dry conditions. As shown in Figure 6 (b) and Table S4, the bold peak at 3360 $cm^{-1}$ corresponds to the characteristic peak of C-OH in alcohol. The peak at 3192$cm^{-1}$ originates from the O-H stretching vibration of carboxylic acid. The two characteristic peaks at 2921 $cm^{-1}$ and 2850 $cm^{-1}$ corresponds to the C-H stretching vibration of alkane. The peaks at 1660 $cm^{-1}$ and 1633 $cm^{-1}$ originate from C=O stretching vibrations. The

signals at 1465 cm$^{-1}$ and 1415 cm$^{-1}$ represent the deformation vibrations of methyl and methylene groups. The peak around

1268 cm$^{-1}$ corresponds to the vibration of nitrate groups in nitrate ester. The results above suggest that the SOA extracts are dominantly composed of carbonyl compounds, carboxylic acid, nitrate ester, and alcohol. This is consistent with previous studies (Huang et al., 2015; Fahnestock et al., 2015).

### 3.3 Factors affecting the formation of SOA and ozone

According to previous studies, ozone concentration was statistically positively correlated with temperature, solar radiation

intensity, and sunshine hours, and was negatively correlated with precipitation, relative humidity (RH), visibility, and wind speed (Wang et al., 2017; Huang et al., 2019; Jaffe and Zhang, 2017). The factors affecting ozone generation included solar radiation intensity, temperature, and precursor concentrations. Figure 7 showed the generation of ozone during the photochemical reactions of three different reaction systems. As shown in Figure 7 (a), for the n-dodecane reaction system, the ozone concentration was Dod-1 > Dod-2. As revealed by Table S2 and Figure S1, under similar $\Delta$VOCs/NO$_X$ ratio and

solar radiation intensity (J(NO$_2$)), the higher temperature would promote the formation of ozone. For 1,3,5-TMB reaction system, the formed ozone concentration followed the order: TMB-2 > TMB-3 > TMB-1. The corresponding $\Delta$VOCs/NO$_X$ ratio were 8.13, 6.12, and 4.48, respectively. The temperature conditions were TMB-1 > TMB-2 > TMB-3. The J(NO$_2$) was similar for the three experiments. With similar solar radiation intensity, the $\Delta$VOCs/NO$_X$ ratio of precursors played a decisive role in the generation of ozone concentration compared to the temperature conditions. For the mixture AVOCs

reaction system, the order of formed ozone concentration was MIX-2 > MIX-1. The corresponding $\Delta$VOCs/NO$_X$ ratio was 7.83 and 8; the J(NO$_2$) was MIX-1 ~ MIX-2. The TMB/Dod ratio were 7 and 6. This indicated that, under similar $\Delta$VOCs/NO$_X$ ratio and J(NO$_2$) conditions, a higher TMB/Dod ratio would promote the formation of ozone. For experiments MIX-2 and MIX-3, the $\Delta$VOCs/NO$_X$ ratio were similar, the temperature conditions were MIX-2 > MIX-3, the J(NO$_2$) was MIX-2 ~ MIX-3, the TMB/Dod ratio were 9.1 and 7. The conditions above indicated that the higher TMB/Dod ratio played a

decisive role in the generation of ozone concentration. Higher temperature and higher $\Delta$VOCs/NO$_X$ (ppbC/ppb) ratio in a separate reaction system will promote the generation of ozone; the relative content of reaction precursors (ppb/ppb) in the mixture system will affect the concentration of ozone, with similar $\Delta$VOCs/NO$_X$ ratio, a higher concentration of 1,3,5-TMB will promote ozone generation.

As shown in Figure 8, for the n-dodecane reaction system, the mass concentration, number concentration, and total

surface of the particles were Dod-1 < Dod-2. According to Figure S1 and Table S2, under similar $\Delta$VOCs/NO$_x$ ratio and J(NO$_2$), the lower temperature would promote the formation of particles. However, the surface mean and number mean diameter of the two experiments were around 100 and 200 nm, this indicated that temperature had little effect on the diameter of the formed particles. For 1.3.5-TMB reaction system, under similar J(NO$_2$), lower temperature and higher $\Delta$VOCs/NO$_x$ ratio would promote the particle formation; temperature and $\Delta$VOCs/NO$_x$ ratio had little effect on the particle

diameters. For the mixture experiments, under similar $\Delta$VOCs/NO$_x$ ratio, compared with J(NO$_2$) and temperature, a higher Dod/TMB ratio would promote the particle formation; similarly, the above conditions had little effect on the particle size.

Lower temperature and higher $\Delta VOCs/NO_X$ (ppbC/ppb) ratio in a separate reaction system will promote the particle formation; the relative content of reaction precursors (ppb/ppb) in the mixture system will affect the formed particles, with similar $\Delta VOCs/NO_X$ ratio, a higher concentration of n-dodecane would promote the generation of particles; reaction
conditions have little effect on the size of the final particle size.

## 4 Atmospheric Implications

Our findings demonstrate that the SOA yield derived from the mixed anthropogenic volatile organic compounds (n-dodecane
+ 1,3,5-TMB) in the presence of HONO is higher than the predicted value. The results of this work further demonstrate the inaccuracy of the SOA yield calculation method for the VOCs mixture, i.e., the simple linear addition of SOA yields from the individual yield of the compound in the VOCs mixture. This calculation method may underestimate or overestimate the SOA production. In this work, the SOA production from the mixed n-dodecane and 1,3,5-TMB is underestimated. In the general case, the SOA yields from the individual compounds should be used with caution when calculating the SOA yields
from the VOCs mixture. Besides, as the representative substances of motor-vehicle exhaust, long-chain alkanes and aromatic hydrocarbons exist in the atmosphere at the same time. The increase in SOA yield after mixing the two kinds of compounds gives us an insight into the SOA yield derived from the vehicular exhaust. Our results indicate that SOA formation needs to be considered more realistically in the atmosphere.

## 5 Conclusions

In summary, a set of photochemical experiments are carried out in a large-scale outdoor smog chamber. The measured SOA mass concentration of the mixture AVOCs (n-dodecane + 1,3,5-TMB) is compared to the predicted SOA mass concentration based on the SOA mass yields of the individual compounds. Results show that the SOA formation from the mixture AVOCs is enhanced. Mass spectra of the SOA particles indicate that interaction occurs between the intermediate products from the two precursors, and the products previously present in the gas phase may enter the particle phase through this inter-reaction.
This could be the main reason for the enhanced SOA production from the mixture AVOCs reaction system. The SOA formation and the ozone formation vary with the $NO_X/VOC$ ratio, the temperature, and the solar radiation intensity.

Further research is needed to study the SOA chemistry from biogenic-biogenic VOC mixtures, biogenic-anthropogenic VOC mixtures, and anthropogenic-anthropogenic VOC mixtures. The interactions between VOC mixtures and the effect on SOA formation are needed to be determined.


*Data availability*. The data used in this study are available upon request from the corresponding author.

*Author contributions*. Junling Li and Hong Li conceived and led the studies. Junling Li, Kun Li, Yan Chen, Hao Zhang, Xin Zhang, and Zhenhai Wu performed chamber simulation and data analysis. Hong Li, Yongchun Liu, Xuezhong Wang, Weigang Wang, and Maofa Ge discussed the results and commented on the paper. Junling Li prepared the article with contributions from all co-authors.

*Competing interests*. The authors declare that they have no conflict of interest.

*Acknowledgements*. This project was supported by the Beijing Municipal Science & Technology Commission (No. Z181100005418015), China Postdoctoral Science Foundation (2019M660752), and LAC/CMA (2019B08). We would like to thank Mr. Bin Liang from Bruker for supporting us in mass spectrometry analysis.

*Financial support*. This project was supported by the Beijing Municipal Science & Technology Commission (No. Z181100005418015), China Postdoctoral Science Foundation (2019M660752), and LAC/CMA (2019B08).

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

**Table 1. Summary of the initial conditions and results of the conducted experiments.**

| Number | Date | Initial Conditions of the Experiments | | | | | | | | | General Results of the Experiments | | | | | |
|---|---|---|---|---|---|---|---|---|---|---|---|---|---|---|---|---|
| | | 1,3,5-TMB (ppb) | $n$-dodecane (ppb) | NO (ppb) | $NO_2$ (ppb) | $NO_X$ (ppb) | Δ VOCs/NOx (ppbC/ppb) | T (at noon) (°C) | $J(NO_2)$ (at noon) ($s^{-1}$) | RH (%) | $O_3$ (ppb) | Mass[a] ($\mu g/m^3$) | Mass Predicted[b] ($\mu g/m^3$) | Mass Corr.[c] ($\mu g/m^3$) | Mass Predicted[d] ($\mu g/m^3$) | Yield[e] (%) |
| Dod-1 | 2019.09.27 | -- | 22 | 50 | 160 | 210 | 1.26 | 37 | 0.0050 | < 2 | 57 | 6.4 | -- | 35.2 | -- | 23.1 |
| Dod-2 | 2019.10.09 | -- | 20 | 77 | 137 | 214 | 1.12 | 34 | 0.0048 | < 2 | 25 | 3.7 | -- | 19.1 | -- | 14.1 |
| TMB-1 (Li et al., 2021) | 2019.09.03 | 105 | -- | 23 | 188 | 211 | 4.48 | 43 | 0.0056 | < 4 | 288 | 2.1 | -- | 7.68 | -- | 1.5 |
| TMB-2 (Li et al., 2021) | 2019.09.25 | 178 | -- | 46 | 151 | 197 | 8.13 | 38 | 0.0053 | < 4 | 772 | 5.1 | -- | 21.11 | -- | 2.4 |
| TMB-3 (Li et al., 2021) | 2019.10.14 | 170 | -- | 68 | 182 | 250 | 6.12 | 30 | 0.0055 | < 5 | 530 | 2.5 | -- | 8.99 | -- | 1.1 |
| MIX-1 | 2019.09.07 | 168 | 28 | 62 | 169 | 231 | 8 | 46 | 0.0058 | < 3 | 358 | 59.3 | 10.3 | 73.5 | 49.9 | -- |
| MIX-2 | 2019.09.21 | 155 | 22 | 58 | 154 | 212 | 7.83 | 39 | 0.0056 | < 2 | 721 | 47.4 | 8.5 | 58.5 | 40.9 | -- |
| MIX-3 | 2019.09.19 | 182 | 20 | 71 | 147 | 218 | 8.61 | 31 | 0.0044 | < 9 | 435 | 11.5 | 8.6 | 14.8 | 40.8 | -- |
| MIX-4 | 2020.08.21 | 251 | 35 | 54 | 158 | 212 | 12.64 | 39 | | < 7 | 999 | 60.2 | 13.7 | 74.6 | 65.8 | -- |
| MIX-5 | 2020.07.14 | 4 h add | 27 | 61 | 157 | 218 | -- | 52 | 0.0051 | < 5 | 289 | 8 | -- | 27.7 | -- | -- |
| MIX-6 | 2020.07.20 | 4 h add | 38 | 58 | 198 | 256 | -- | 43 | 0.0057 | < 4 | 276 | 6 | -- | 20.8 | -- | -- |
| MIX-7 | 2020.07.24 | 1 h add | 39 | 56 | 207 | 263 | -- | 42 | 0.0058 | < 6 | 440 | 2.3 | -- | 7.9 | -- | -- |
| MIX-8 | 2020.07.22 | 227 | 1h add | 48 | 167 | 216 | -- | 43 | 0.0051 | < 5 | 335 | 4.5 | -- | 15.6 | -- | -- |

a: the mass here is the measured value with the SMPS; the density of the formed SOA derived from 1,3,5-TMB is assumed to be 1.4 $g/cm^3$ (Zhang et al., 2016; Nakao et al., 2013); the density of the formed SOA derived from $n$-dodecane is assumed to be 1.06 $g/cm^3$ (Lim and Ziemann, 2009; Li et al., 2017a); the density of the formed SOA derived from the mixed AVOCs is assumed to be 1.23 $g/cm^3$.

b: the predicted mass here is based on the yield that the particle and vapor wall-loss are not considered.

c: the corrected mass here is calculated after taking particle and vapor wall loss into account.

d: the predicted mass here is based on the yield that the particle and vapor wall-loss are considered.

e: the SOA yield here is calculated after taking particle and vapor wall loss into account.

**Table 2. Representative identified mass spectral peaks, molecular formulas, molecular weights, and relative intensity of n-dodecane, 1,3,5-TMB, and mixture AVOCs-derived SOA.**

| Molecular Formula | M+H | M+Na | MIX-Relative Intensity ($\times 10^{-3}$) | TMB-Relative Intensity ($\times 10^{-3}$) | n-dodecane Relative Intensity ($\times 10^{-3}$) |
|---|---|---|---|---|---|
| $C_9H_{14}O_3$ | 171.099 | | 1.41 | 1.73 | 0.733 |
| $C_9H_{17}NO_3$ | | 210.111 | 0.484 | 2.38 | 0.864 |
| $C_{11}H_{18}O_4$ | 215.126 | | 0.641 | 0.0563 | 0.0489 |
| $C_{14}H_{22}O_3$ | 239.166 | | 0.217 | 0.0526 | 0.0489 |
| $C_{14}H_{20}O_2$ | | 243.134 | 0.113 | 0.0966 | 0.110 |
| $C_{11}H_{22}NO_5$ | 249.158 | | 0.489 | 0.0583 | 0.0718 |
| $C_{14}H_{26}O_2$ | | 249.183 | 0.503 | 0.0495 | 0.0459 |
| $C_{13}H_{25}NO_2$ | | 250.177 | 1.22 | 1.79 | 1.69 |
| $C_8H_{12}O_9$ | 253.056 | | 0.220 | 0.0308 | 0.0109 |
| $C_{11}H_{22}O_5$ | | 257.135 | 0.535 | 0.0265 | 0.0223 |
| $C_{12}H_{20}O_6$ | 261.131 | | 0.387 | 0.284 | 0.393 |
| $C_{15}H_{28}O_2$ | | 263.199 | 0.250 | 0.0808 | 0.0427 |
| $C_{14}H_{24}O_5$ | 273.167 | | 0.799 | 0.0872 | 0.0643 |
| $C_{16}H_{22}O_4$ | 279.159 | | 0.728 | 0.139 | 0.0724 |
| $C_{14}H_{26}O_4$ | | 281.172 | 0.305 | 0.0685 | 0.0427 |
| $C_{18}H_{28}O$ | | 283.207 | 1.14 | 1.83 | 1.46 |
| $C_{13}H_{22}NO_6$ | 289.153 | | 0.890 | 0.0513 | 0.0565 |
| $C_{16}H_{22}O_4$ | | 301.146 | 4.77 | 1.09 | 1.41 |
| $C_{18}H_{34}O_2$ | | 305.263 | 3.26 | 1.22 | 0.004.4 |
| $C_{16}H_{30}O_4$ | | 309.202 | 1.28 | 0.149 | 0.135 |
| $C_{18}H_{28}O_3$ | | 315.194 | 0.936 | 1.25 | 1.26 |
| $C_{16}H_{24}O_5$ | | 319.151 | 2.09 | 0.0290 | 0.0197 |
| $C_{20}H_{34}O_2$ | | 329.246 | 0.319 | 0.0927 | 0.133 |
| $C_{19}H_{38}O_4$ | | 353.267 | 1.42 | 1.46 | 2.42 |
| $C_{24}H_{38}O_4$ | | 413.266 | 2.69 | 1.39 | 1.74 |
| $C_{20}H_{34}O_8$ | | 425.214 | 0.297 | 0.395 | 0.0333 |
| $C_{24}H_{36}NaO_5$ | | 427.245 | 0.107 | 0.108 | 0.0393 |
| $C_{27}H_{48}O_8$ | | 523.325 | 0.183 | 1.54 | 1.33 |
| $C_{30}H_{60}NO_6$ | | 553.459 | 3.08 | 0.145 | 3.32 |
| $C_{28}H_{48}O_{10}$ | | 567.307 | 0.272 | 0.0195 | 0.0215 |
| $C_{29}H_{48}O_{10}$ | | 579.296 | 1.54 | 0.0150 | 0.0167 |
| $C_{35}H_{68}O_{10}$ | | 639.480 | 1.2 | 0.109 | 0.134 |
| $C_{41}H_{60}NO_6$ | | 685.434 | 2.01 | 0.209 | 2.11 |


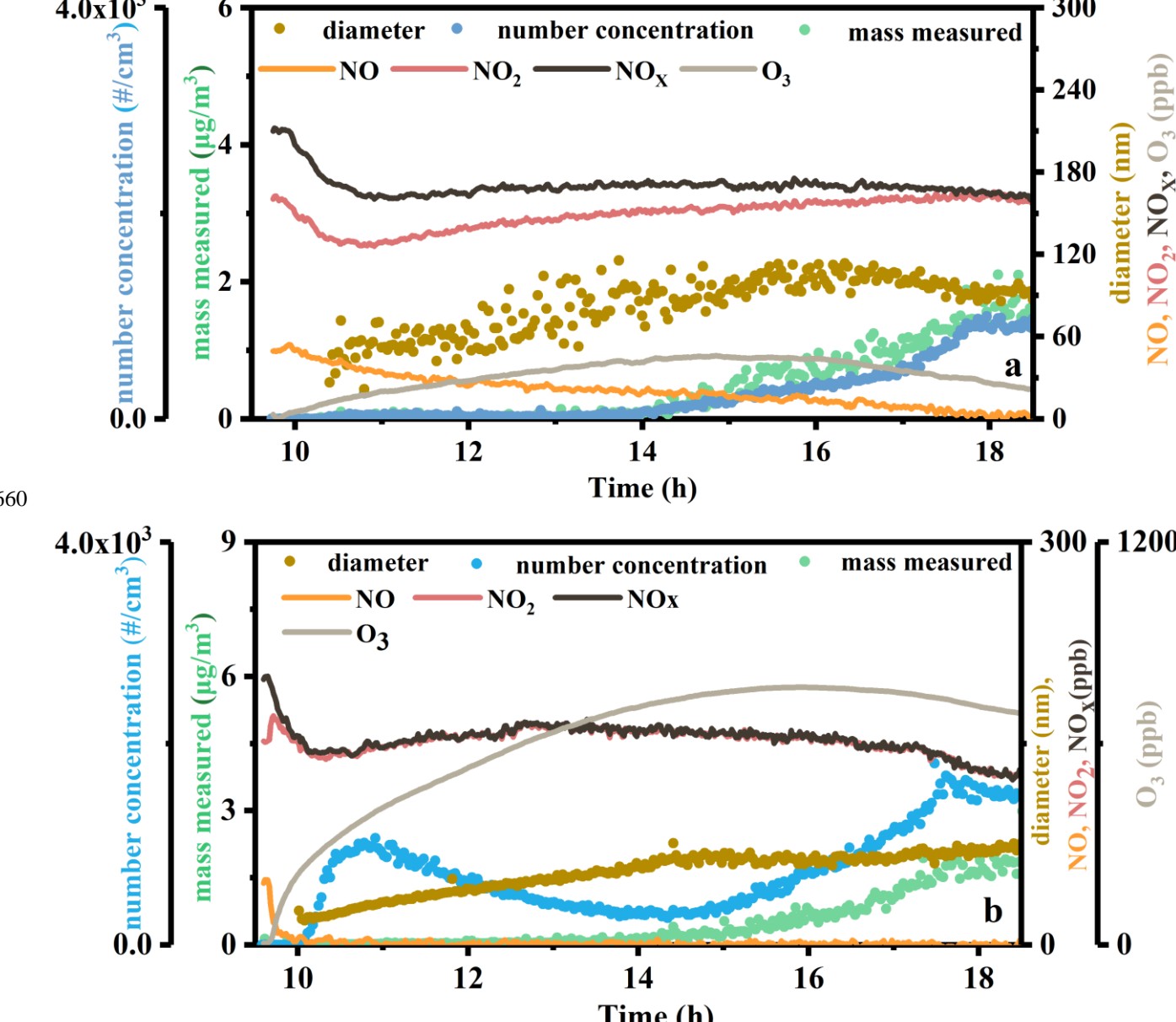

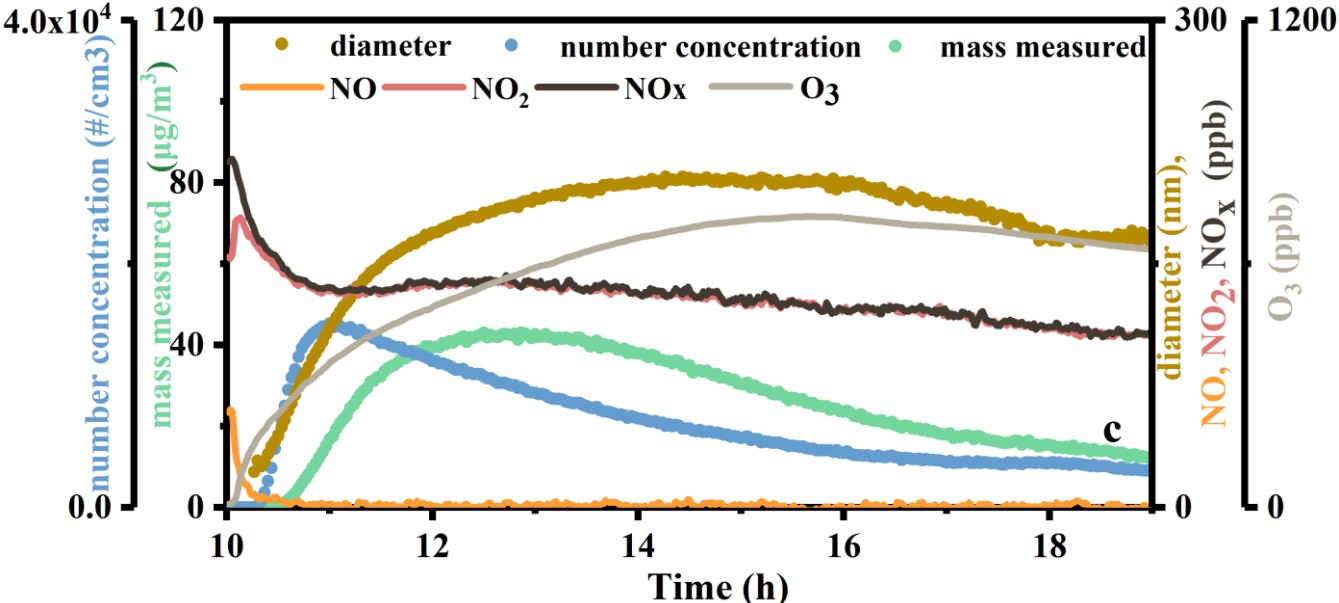

**Figure 1.** Reaction profiles of photooxidation of n-dodecane (a), 1,3,5-TMB (b), and mixture AVOCs (c) under $NO_X$ conditions in summer. The concentrations of mass and number concentration of particles are shown on the left axes, while the diameter of particles and concentrations of NO, $NO_2$, $NO_x$, and $O_3$ are shown on the right axes.

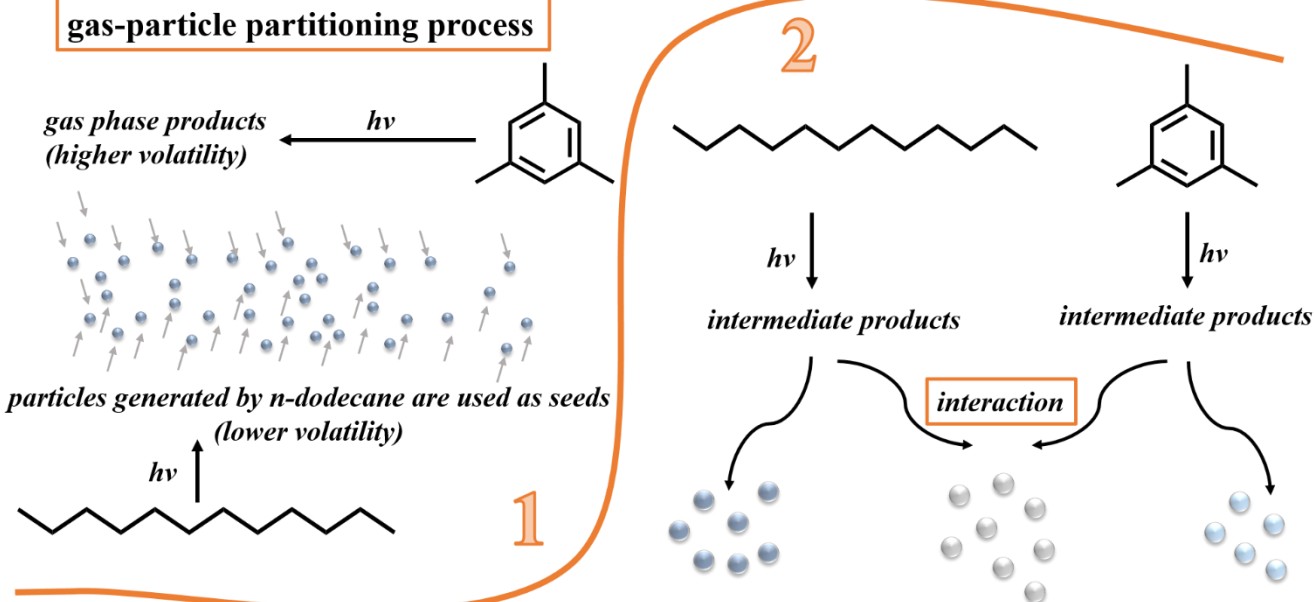

**Figure 2.** The possible conjectures of the reaction processes.

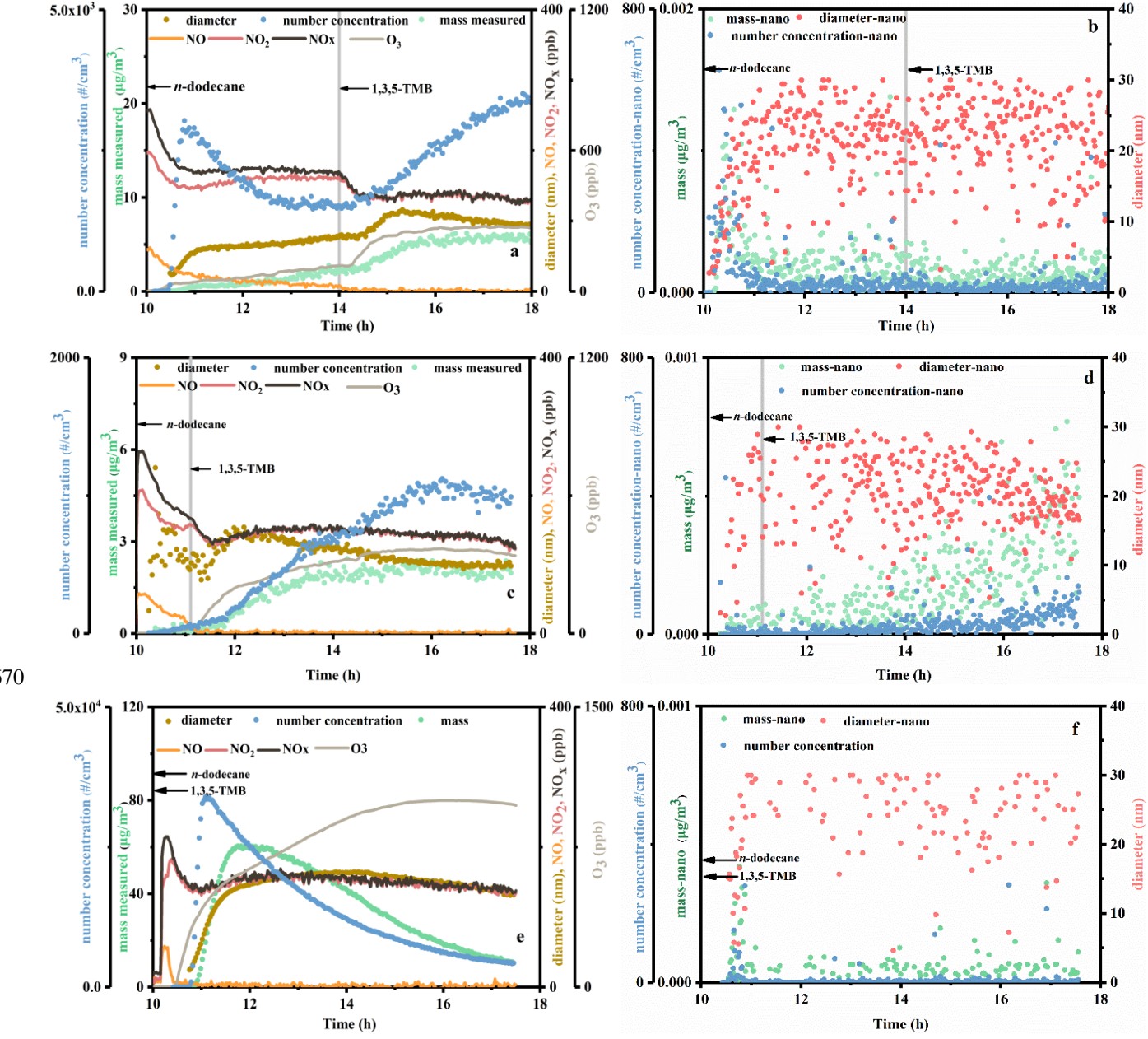

**Figure 3.** (a) reaction profiles of experiment MIX-6; (b) time series of particles in the size range of 0-40 nm for MIX-6; (c) reaction profiles of experiment MIX-7; (d) time series of particles in the size range of 0-40 nm for MIX-7; (e) reaction profiles of experiment MIX-4; (f) time series of particles in the size range of 0-40 nm for MIX-4. The concentrations of mass and number concentration of particles are shown on the left axes, while the diameter of particles and concentrations of NO, NO₂, NOₓ, and O₃ are shown on the right axes. The vertical gray lines in the figures refer to the time that 1,3,5-TMB was added.



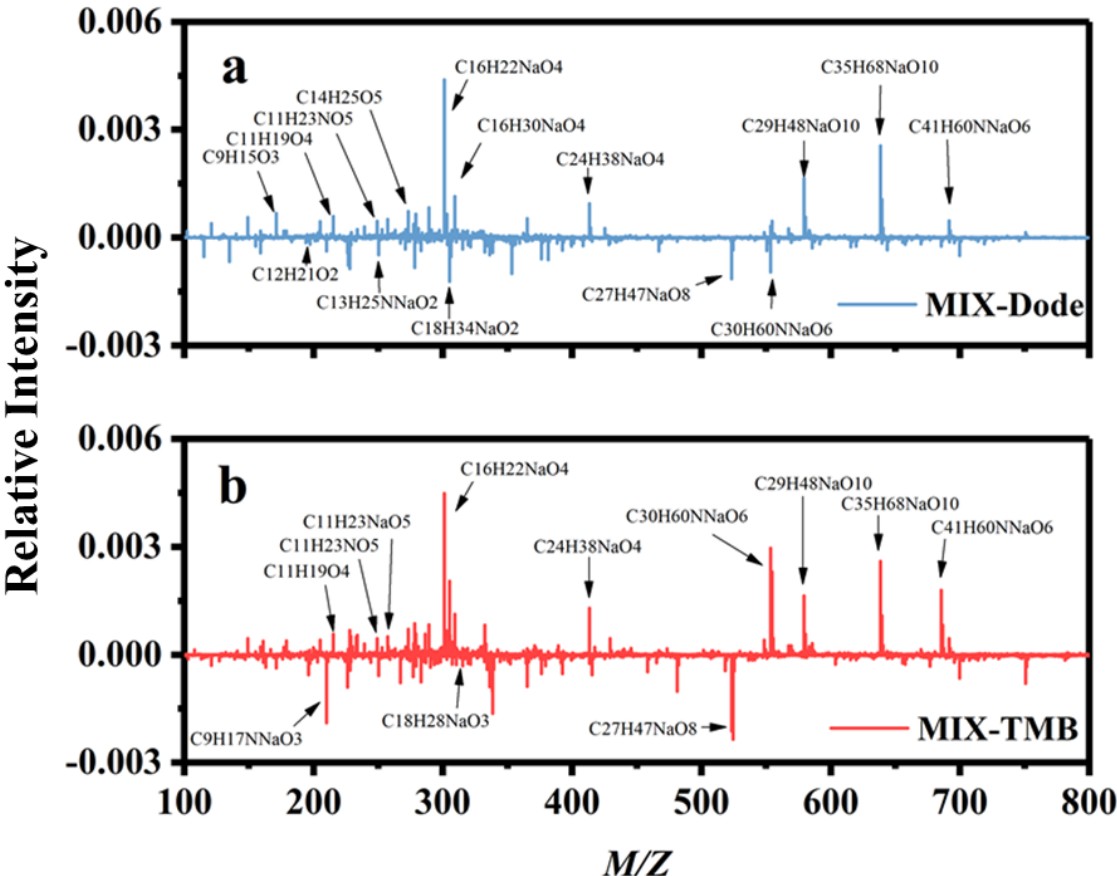

**Figure 4. Results of mass spectra difference in (a) mixed AVOCs SOA minus *n*-dodecane SOA, and (b) mixed AVOCs SOA minus 1,3,5-TMB SOA. The Y-axis is the relative intensity normalized by dividing by the total signal strength of the mass spectra.**


**Figure 5. Proposed reaction mechanism of mixture AVOCs system in the presence of NO$_X$ (R1 and R2 are alkyl groups or aromatic hydrocarbon groups).**

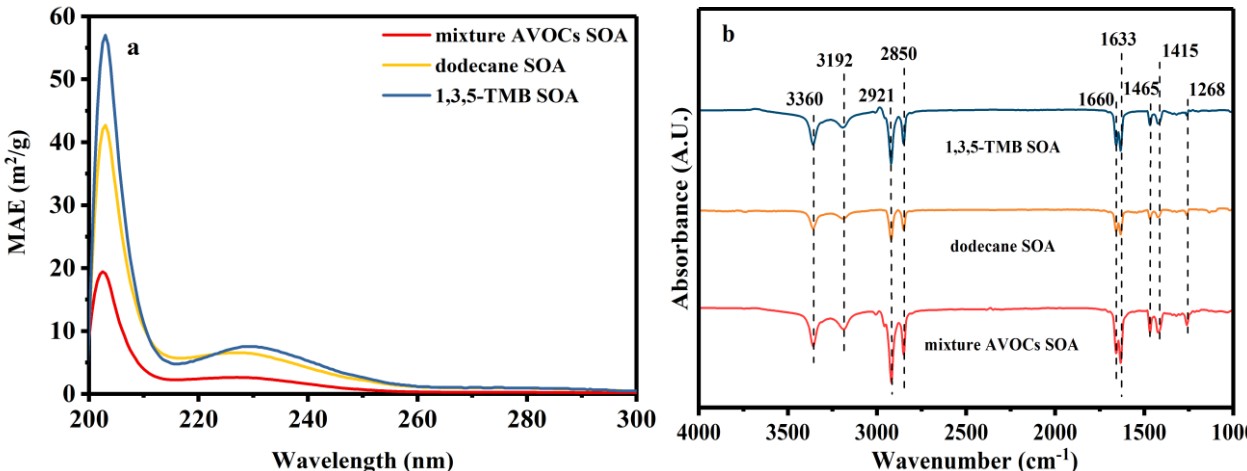


**Figure 6. (a)** UV-Vis spectra (MAE) of the *n*-dodecane, 1,3,5-TMB, and mixture AVOCs SOA. As the absorbance at wavelengths >300 nm is negligible, we only show the range from 200 to 300 nm. **(b)** ATR-IR spectra for the *n*-dodecane, 1,3,5-TMB, and mixture AVOCs SOA by using a background spectrum obtained without a sample as the reference.

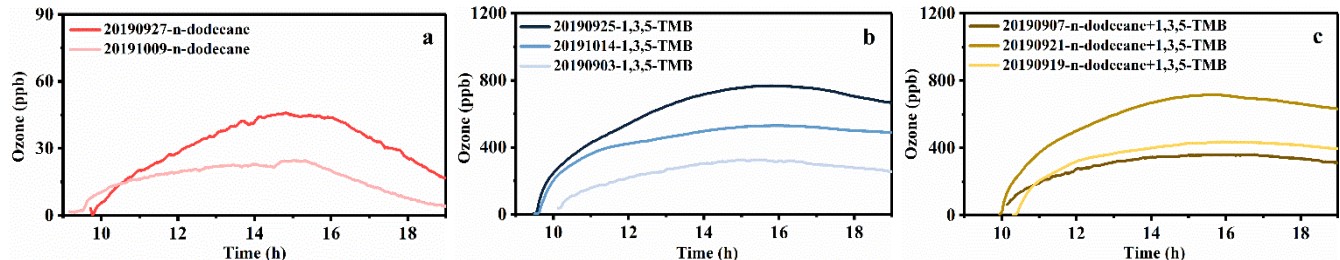


**Figure 7.** Ozone formation during the photochemical processes of the *n*-dodecane (a); 1,3,5-TMB (b); and mixture AVOCs (c).

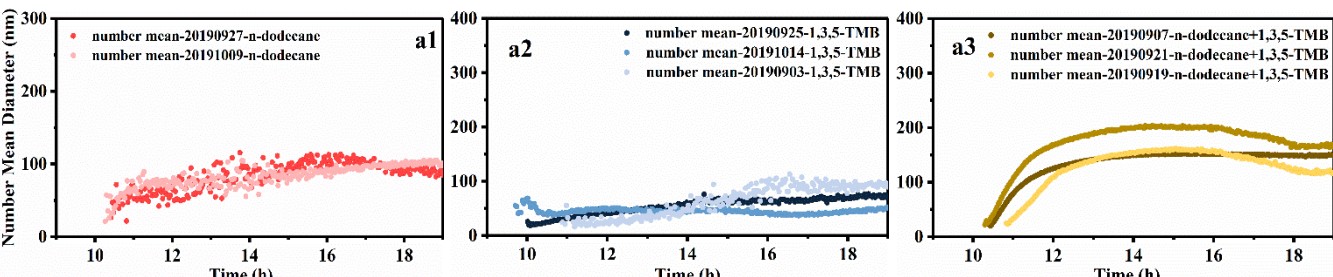

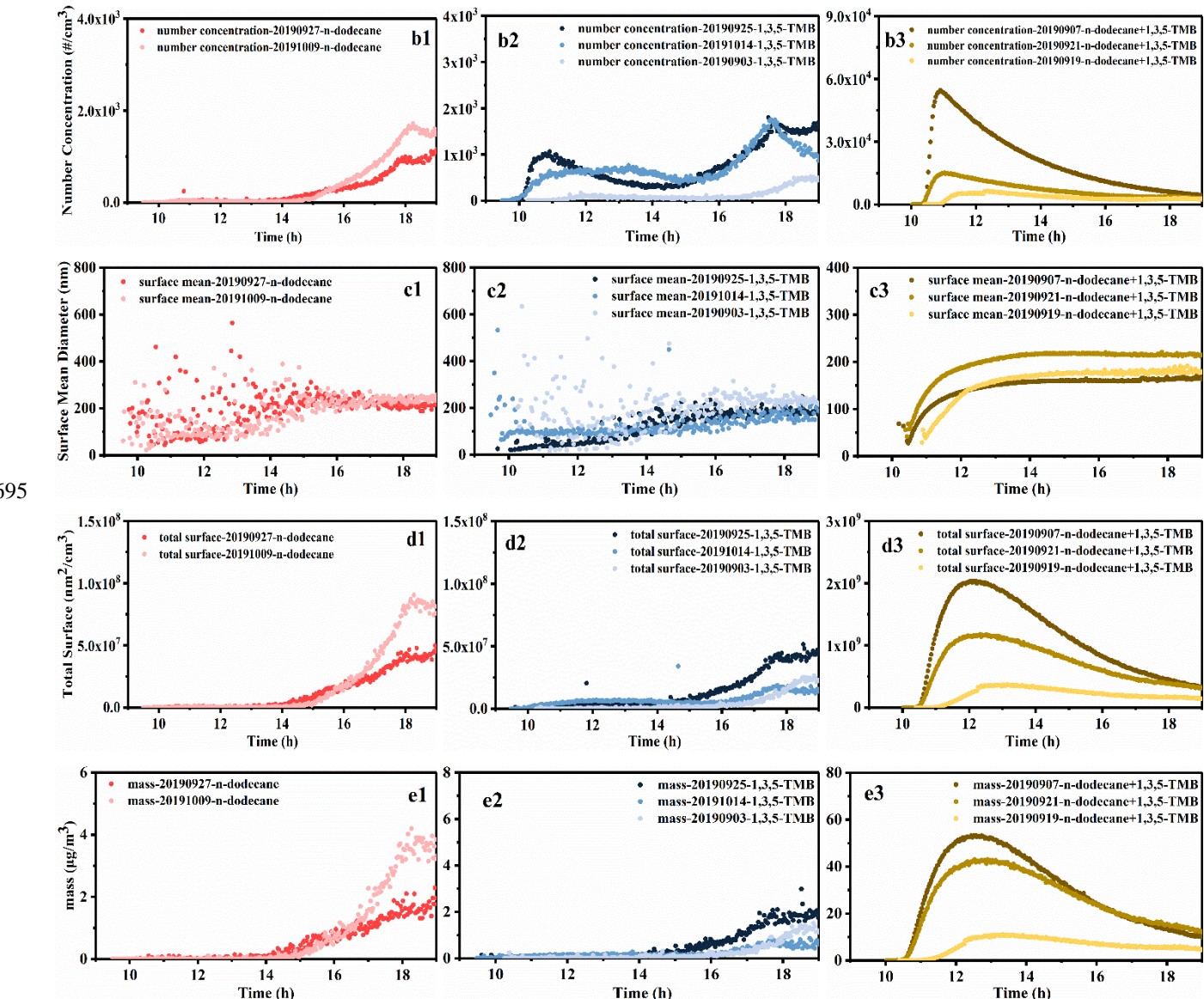

**Figure 8**. **The formation and evolution of particles during the photochemical reactions in summer. The number mean diameter of particles derived from *n*-dodecane (a1), 1,3,5-TMB (a2), mixture AVOCs (a3); the number concentration of particles derived from *n*-dodecane (b1), 1,3,5-TMB (b2), mixture AVOCs (b3); the surface mean diameter of particles derived from *n*-dodecane (c1), 1,3,5-TMB (c2), mixture AVOCs (c3); the total surface of particles derived from *n*-dodecane (d1), 1,3,5-TMB (d2), mixture AVOCs (d3); the mass concentration of particles derived from *n*-dodecane (e1), 1,3,5-TMB (e2), mixture AVOCs (e3).**


