# Peer review of "Enhanced secondary organic aerosol formation from the photo-oxidation of mixed anthropogenic volatile organic compounds"

_Atmospheric Chemistry and Physics, 2020_

## Referee Comment (RC1) · Anonymous Referee #1 · 30 Dec 2020

General Comments:

This manuscript presents the SOA production from individual anthropogenic VOC precursors (i.e., n-dodecane and 1,3,5-trimethylbenzene) versus the system of mixed VOCs. Enhanced SOA yields were observed with mixed VOC precursors compared to the linear addition of SOA yields derived from individual VOC precursors, indicating the significance of chemical interactions between intermediate products from these two precursors. Overall, this study provides useful information and highlights the complexity of SOA chemistry in the mixture of VOCs representative of real atmosphere. One major comment is that although ESI-TOF-MS data were presented (in Table 2), the

molecular composition of SOA unique for each system was not discussed in detail to probe the underlying chemical processes. More in-depth discussion is required. The difference between measured m/z and theoretical m/z of proposed molecular formula should also be reported for appropriate QA/QC of accurate mass fittings.

Specific Comments:

(1) Line 166: $5.0 \times 10^7$ nmˆ2/cmˆ3 (please correct the superscripts here)

(2) Line 219: What are the repeating units (i.e., monomers) of oligomers detected in the range of m/z 500-700? Are they related to the reactive intermediates of individual VOC precursors? More information is needed to directly support the chemical interactions between two precursors.

(3) Line 244-247 and 301-303: To determine the potential functional groups in SOA extracts, it would be best to acquire complementary IR spectra. It seems premature to reach these conclusions based on the UV-Vis spectra shown in Figure 5.

(4) For Figure 5, the absorbance is proportional to the concentration of SOA extracts. What are the mass concentrations of SOA solutions for samples presented in Figure 5? It would be better to present the mass absorption coefficients (MACs) to understand their light absorption properties.

(5) Line 299: interaction "occurs"

---

## Referee Comment (RC2) · Anonymous Referee #2 · 12 Feb 2021

General Comments The reviewed study investigates combining precursors in SOA experiments and the effect of the combination in its ability to produce SOA. Interestingly the authors observe that combining an alkane and aromatic precursor in their experiments results in an SOA yield that is greater than the weighted average of each precursor's individual SOA yield. This could be from a change in partitioning or from new chemical pathways that are activated only in the combination of dodecane and TMD. The authors state the cause is from the occurring chemistry, and not just partitioning. This is justified by looking at molecular species with ESI-TOF-MS observed in in experiments as well as the UV-Vis absorbance of the resulting SOA extract. The authors don't make the strongest case for what specifically changes in the occurring chemistry

to generate more SOA, but they do point to measurable differences, such as unique ESI-TOF-MS peaks in the mixed experiments. Overall, the results are interesting and worthy of publication. Specific Comments 1. In Figure 1 a and b you show that the corrected mass is substantially greater than the experimentally measured mass, by at least an order of magnitude it appears. This is concerning as there is undoubtedly some error in the correction that is used. Could the authors please estimate the error associated with the mass correction they use and how that propagates into the "mass corrected" values in Figure 1. 2. In Figure 5 it is difficult to see the difference between peaks because of the x-axis scale. It would be more informative to make the x-axis from 200 – 300 nm, and simply note in the caption that absorbance at wavelengths >300 nm is negligible. 3. It seems erroneous to say the observed peaks around 205 nm are strictly due to carboxyl. Given the presence of NO in your experiments, you will have nitrate functionality in your products. Nitrate absorbs strongly around 210 nm, but this can be shifted from neighboring functional groups, etc. 4. Based on figure 5, you cannot say (line 243) that the mixed AVOCS forms more carboxyl compounds. Because the mixed experiments contains more aerosol mass, you would expect the absorbance to be greater even with the same yield of carboxyls (or nitrate, see comment 3) relative to the non-mixed experiments. 5. The authors note a handful of peaks from ESI that only appear in mixed experiments, but in Figure 4 it is not especially clear that there are many peaks in panel C that are not in panel A and B. Does the ESI have similar sensitivity for all observed species? Or can sensitivity vary greatly between species? Please address this, it would help the reader to interpret figure 4. 6. Also, as it is difficult to see the magnitude of the peaks unique to the mixed experiments in panel C - is it possible that the unique peaks are present in unmixed experiments but just under detectable limits, and upon producing greater aerosol in the mixed experiments, the unique peaks were elevated to above detectable limits? Perhaps a useful way to address this would be give the intensities of the unique peak relative to the larges peak of the spectrum, in addition to addressing comment 5. Technical corrections. 6. Line 84: Add "The" before "OH precursor" 7. Line 84 and 86: Correct subscripts in molecular formulas. 8. Line

219: m/z is commonly italicized, m/z, this should be corrected throughout document.

9. Line 243: "formes" should be forms

---

## Author Comment (AC1) · 20 Mar 2021

**Response to the comments of Reviewer #1**

*This manuscript presents the SOA production from individual anthropogenic VOC precursors (i.e., n-dodecane and 1,3,5-trimethylbenzene) versus the system of mixed VOCs. Enhanced SOA yields were observed with mixed VOC precursors compared to the linear addition of SOA yields derived from individual VOC precursors, indicating the significance of chemical interactions between intermediate products from these two precursors. Overall, this study provides useful information and highlights the complexity of SOA chemistry in the mixture of VOCs representative of real atmosphere.*

Response: We thank Anonymous Referee #1 for the review and the positive evaluation of our manuscript. We have fully considered the comments and responded to these comments below in blue text. The revisions in the manuscript are highlighted in yellow color. The response and changes are listed below.

*Major Comments:*

1. *One major comment is that although ESI-TOF-MS data were presented (in Table 2), the molecular composition of SOA unique for each system was not discussed in detail to probe the underlying chemical processes. More in-depth discussion is required.*

    We thank the reviewer for pointing this out. The discussion on the underlying chemical processes and the proposed reaction mechanism of the mixture AVOCs system are added in the manuscript.

    **Page 8, line 232-244**: "The gas phase products of OH-initiated oxidation of 1,3,5-TMB in the presence of $NO_x$ are mainly 3,5-dimethyl benzaldehyde ($C_9H_{10}O$), 3,5-dimethylbenzoic acid ($C_9H_{10}O_2$), 2-methyl-4-oxo-2-pentenal ($C_6H_8O_2$), 2-methyl-4-oxo-2-pentenoic acid ($C_6H_8O_3$), 2,4,6-trimethylphenol ($C_9H_{12}O$), and 3,5-dimethyl-2-furanone ($C_6H_8O_2$) (Huang et al., 2015), which contain carbonyl or hydroxyl groups that are formed within 1h photochemical reaction. The intermediate products of OH-initiated oxidation of n-dodecane in the presence of $NO_x$ are also compounds containing carbonyl and hydroxyl groups, and more alcohol can be formed due to $RO_2$ + NO reaction compared to low $NO_X$ condition (Fahnestock et al., 2015). These compounds tend to undergo acetal reaction and/or esterification reaction in the particle phase. When the photochemical reaction is initiated, the intermediate products produced by 1,3,5-TMB and n-dodecane exist in the

same reaction system, acetal and esterification reactions are more likely to occur in the particle phase due to higher concentration of aldehydes, ketones, alcohols, and carboxylic acids. The proposed reaction mechanism of the mixture AVOCs system is shown in Figure 5. As an example, the $C_{16}H_{24}O_5$, which has a much higher intensity in the mixed AVOCs system (as shown in Table 2 and discussed above), might be an ester from the reaction of an acid and an alcohol from 1,3,5-TMB and n-dodecane, respectively."

Figure 5. Proposed reaction mechanism of mixture AVOCs system in the presence of $NO_X$. (**Page 26, line 645-646**)

2. *The difference between measured m/z and theoretical m/z of proposed molecular formula should also be reported for appropriate QA/QC of accurate mass fittings.*

The ESI-ToF-MS used in this study has very high mass resolution, and the mass error between measure $m/z$ and theoretical $m/z$ of the proposed formula was below 5 ppm. We have added the "Quality Assurance and Quality Control" section in the Supporting Information.

**Supporting Information, Page 2, line 28-34**: "The collections and analysis of SOA samples were under strict quality control. The Teflon tube in the sampling device was purged with zero air before sampling, and the membrane holder was cleaned with methanol. Glassware used in the experiments was washed with water and methanol, and then dried under high temperature conditions. Before each injection, the micro syringes were cleaned with methanol to prevent cross-contamination among different samples. The blank PTFE filter analysis showed that there was no serious contamination. The typical mass resolving power of the applied ESI-ToF-MS was > 80000 at $m/z$ 1222. The absolute mass error between the measured $m/z$ and theoretical $m/z$ was below 5 ppm."

*Specific Comments:*

3. *Line 166: 5.0 × 10ˆ7 nmˆ2/cmˆ3 (please correct the superscripts here)*

   We have corrected this in the manuscript.

   **Page 6, line 173**: "5.0 × 107 nm2/cm3 → $5.0 \times 10^7$ $nm^2/cm^3$".

4. *Line 219: What are the repeating units (i.e., monomers) of oligomers detected in the range of m/z 500-700? Are they related to the reactive intermediates of individual VOC precursors? More information is needed to directly support the chemical interactions between two precursors.*

   As each precursor can generate intermediate products with smaller carbon numbers, and these intermediate products can oligomerize, the reactions are very complex. The compounds at $m/z$ 500-700 have carbon numbers of $C_{27}$-$C_{41}$. Concerning the high carbon numbers of these compounds and the complexity of the reaction system, it is infeasible to determine the monomers. Nevertheless, we believe our response to the 1st comment has provided more information to support the chemical interactions between two precursors.

5. *Line 244-247 and 301-303: To determine the potential functional groups in SOA extracts, it would be best to acquire complementary IR spectra. It seems premature to reach these conclusions based*

*on the UV-Vis spectra shown in Figure 5.*

We thank the reviewer for the helpful suggestion. We have added the ATR-IR analysis in the manuscript. In Section 2.1, we added:

**Page 3-4, line 96-100**: "The Attenuated Total Internal Reflection Infrared (ATR-IR) analysis was applied to determine the potential functional groups in SOA extracts; an FTIR spectrometer (Bruker, Tensor 27) equipped with a RT-DLaTGs detector was used. The SOA extracts were deposited and dried directly on the Diamant crystal of an ATR-IR cell. The spectra of the dry SOA extracts were recorded by using a background spectrum obtained with no samples as the reference (100 scans, 2.4 cm$^{-1}$ resolution)."

In Section 3.2 "Light absorption of secondary organic aerosol", we added the following contents:

**Page 9, line 259-267**: "To further determine the potential functional groups in SOA extracts, ATR-IR spectra were acquired (Figure 6b). In order to eliminate the influence of water, experiments were conducted under dry conditions. As shown in Figure 6 (b) and Table S3, the bold peak at 3360 cm$^{-1}$ corresponds to the characteristic peak of C-OH in alcohol. The peak at 3192cm$^{-1}$ originates from O-H stretching vibration of carboxylic acid. The two characteristic peaks at 2921 cm$^{-1}$ and 2850 cm$^{-1}$ corresponds to the C-H stretching vibration of alkane. The peaks at 1660 cm$^{-1}$ and 1633 cm$^{-1}$ originate from C=O stretching vibrations. The signal at 1465 cm$^{-1}$ and 1415 cm$^{-1}$ represent the deformation vibrations of methyl and methylene groups. The peak around 1268 cm$^{-1}$ corresponds to the vibration of nitrate groups in nitrate ester. The results above suggest that the SOA extracts are dominantly composed of carbonyl compounds, carboxylic acid, nitrate ester, and alcohol. This is consistent with previous studies (Huang et al., 2015; Fahnestock et al., 2015)."

[Figure]

Figure 6. (a) UV-Vis spectra (MAE) of the *n*-dodecane, 1,3,5-TMB, and mixture AVOCs SOA. As the absorbance at wavelengths >300 nm is negligible, we only show the range from 200 to 300 nm. (b) ATR-IR spectra for the *n*-dodecane, 1,3,5-TMB, and mixture AVOCs SOA by using a background spectrum obtained without a sample as the reference. (**Page 27, line 650-652**)

Table S3. List of functional groups in SOA extracts. (**Page 4, line 70, Supporting Information**)

| Functional groups | Wavenumber (cm$^{-1}$) |
|---|---|
| C-OH in alcohol | 3360 |
| O-H stretching vibration of carboxylic acid | 3192 |
| C-H stretching vibration of alkane | 2921, 2850 |
| C=O stretching vibrations | 1660, 1633 |
| deformation vibrations of methyl and methylene groups | 1465, 1415 |
| nitrate groups in nitrate ester | 1268 |

6. *For Figure 5, the absorbance is proportional to the concentration of SOA extracts. What are the mass concentrations of SOA solutions for samples presented in Figure 5? It would be better to present the mass absorption coefficients (MACs) to understand their light absorption properties.*

The mass concentrations of SOA solutions for samples presented in Figure 5 (now is Figure 6a) are 0.03626 g/m$^3$ (dodecane SOA), 0.0235 g/m$^3$ (1,3,5-TMB SOA), and 0.1127 g/m$^3$ (mixture AVOCs SOA), respectively. To better present the light absorption properties of SOA extracts, the mass absorption efficiency (MAE) has been added in the manuscript.

**Page 8-9, line 251-258**: "Based on the light absorption spectra, the mass absorption efficiency (MAE, m$^2$/g) of the SOA in the extracts is calculated using the following equation (Chen et al., 2016):

$$MAE(\lambda) = ln(10)\, Abs\,(\lambda)/C_{OM} \qquad (7)$$

where *Abs(λ)* is the light absorption coefficient (m$^{-1}$), and $C_{OM}$ is the SOA mass concentrations in

the extracts. The MAE of the SOA extracts in Figure 6 (a) was calculated from 200 to 300 nm. The MAE at 205 nm were in the order: 1,3,5-TMB SOA (56.8 m$^2$/g) > dodecane SOA (42.5 m$^2$/g) > mixture AVOCs SOA (19.5 m$^2$/g). The MAE in the 210-250 nm band also show the same pattern. This indicates that the SOA generated by the mixture AVOCs contains less light-absorbing substance per unit mass relative to dodecane SOA and 1,3,5-TMB SOA."

7. *Line 299: interaction "occurs".*

**Page 10, line 317**: "occur" → "occurs".

---

## Author Comment (AC2) · 20 Mar 2021

**Response to the comments of Reviewer #2**

*The reviewed study investigates combining precursors in SOA experiments and the effect of the combination in its ability to produce SOA. Interestingly the authors observe that combining an alkane and aromatic precursor in their experiments results in an SOA yield that is greater than the weighted average of each precursor's individual SOA yield. This could be from a change in partitioning or from new chemical pathways that are activated only in the combination of dodecane and TMD. The authors state the cause is from the occurring chemistry, and not just partitioning. This is justified by looking at molecular species with ESI-TOF-MS observed in in experiments as well as the UV-Vis absorbance of the resulting SOA extract. The authors don't make the strongest case for what specifically changes in the occurring chemistry to generate more SOA, but they do point to measurable differences, such as unique ESI-TOF-MS peaks in the mixed experiments. Overall, the results are interesting and worthy of publication.*

Response: We thank Anonymous Referee #2 for the review and the positive evaluation of our manuscript. We have fully considered the comments and responded to these comments below in blue text. The revisions in the manuscript are highlighted in yellow color. The response and changes are listed below.

*Specific Comments:*

1. *In Figure 1 a and b you show that the corrected mass is substantially greater than the experimentally measured mass, by at least an order of magnitude it appears. This is concerning as there is undoubtedly some error in the correction that is used. Could the authors please estimate the error associated with the mass correction they use and how that propagates into the "mass corrected" values in Figure 1.*

   The uncertainty of the mass correction is about ±11.2%, and we have added this in the main text.

   **Page 4, line 124-125**: "The uncertainty of the mass correction here is about ±11.2% (see Supporting Information for details)"

   **Supporting Information, Page 2-3, Line 44-57**: "The error associated with the mass

correction here is mainly from the gas phase. For the gas phase wall-loss correction, the gas-particle partitioning timescale ($\bar{\tau}_{g-p}$) and the vapor wall-loss timescale ($\bar{\tau}_{g-w}$) are expressed as the following equation:

$$\bar{\tau}_{g-p} = \frac{1}{2\pi\bar{N}_p\bar{D}_pD_{gas}\bar{F}_{FS}}$$

$$\bar{\tau}_{g-w} = \frac{1}{k_w}$$

$$k_w = \left(\frac{A}{V}\right)\frac{a_w\frac{\bar{c}}{4}}{1.0+\frac{\pi}{2}[\frac{a_w\bar{c}}{4(k_eD_{gas})^{0.5}}]}$$

For $\bar{\tau}_{g-w}$, the parameters in the formula are fixed values, so only $\bar{\tau}_{g-p}$ is considered here. The two variables $N_p$ and $D_p$ in $\bar{\tau}_{g-p}$ are independent and the uncertainty of them are 10% and 1%, respectively. Considering the propagation of uncertainty, we can obtain the variance formula:

$$\sigma_\tau^2 = \sigma_N^2\left(\frac{\partial\tau}{\partial N}\right)^2 + \sigma_D^2\left(\frac{\partial\tau}{\partial D}\right)^2$$

Then the uncertainty of $\bar{\tau}_{g-p}$ can be calculated as:

$$\frac{\sigma_\tau}{\tau} = \sqrt{\left(\frac{\sigma_N}{N}\right)^2 + \left(\frac{\sigma_D}{D}\right)^2} = \sqrt{0.1^2 + 0.01^2} = \sqrt{0.0101} = 10.0\%$$

The uncertainty of $\bar{\tau}_{g-p}$ is $\pm$ 10.0%, the measurement uncertainty of the precursors concentration by TD-GC applied in this work is about $\pm$ 5%, this resulting in the final uncertainty of mass correction to be about $\pm11.2\%$."

2. *In Figure 5 it is difficult to see the difference between peaks because of the x-axis scale. It would be more informative to make the x-axis from 200 – 300 nm, and simply note in the caption that absorbance at wavelengths >300 nm is negligible.*

Thank you for this suggestion. We have updated Figure 5 as Figure 6(a), and the figure caption is also updated.

[Figure]

Figure 6. (a) UV-Vis spectra (MAE) of the n-dodecane, 1,3,5-TMB, and mixture AVOCs SOA. As the absorbance at wavelengths >300 nm is negligible, we only show the range from 200 to 300 nm. (**Page 27, line 650-652**)

3.   *It seems erroneous to say the observed peaks around 205 nm are strictly due to carboxyl. Given the presence of NO in your experiments, you will have nitrate functionality in your products. Nitrate absorbs strongly around 210 nm, but this can be shifted from neighboring functional groups, etc.*

We have modified the description in the manuscript. To further determine the potential functional groups in SOA extracts, the IR spectra are acquired by the Attenuated Total Internal Reflection Infrared (ATR-IR) analysis. We have added this in the manuscript.

**Page 9, line 259-267**: "To further determine the potential functional groups in SOA extracts, ATR-IR spectra were acquired (Figure 6b). In order to eliminate the influence of water, experiments were conducted under dry conditions. As shown in Figure 6 (b) and Table S3, the bold peak at 3360 $cm^{-1}$ corresponds to the characteristic peak of C-OH in alcohol. The peak at 3192$cm^{-1}$ originates from O-H stretching vibration of carboxylic acid. The two characteristic peaks at 2921 $cm^{-1}$ and 2850 $cm^{-1}$ corresponds to the C-H stretching vibration of alkane. The peaks at 1660 $cm^{-1}$ and 1633 $cm^{-1}$ originate from C=O stretching vibrations. The signal at 1465 $cm^{-1}$ and 1415 $cm^{-1}$ represent the deformation vibrations of methyl and methylene groups. The peak around 1268 $cm^{-1}$ corresponds to the vibration of nitrate groups in nitrate ester. The results above suggest that the SOA extracts are dominantly composed of carbonyl compounds, carboxylic acid, nitrate ester, and alcohol. This is consistent with previous studies (Huang et al., 2015; Fahnestock et al., 2015)."

[Figure]

Figure 6. (b) ATR-IR spectra for the n-dodecane, 1,3,5-TMB, and mixture AVOCs SOA by using a background spectrum obtained without a sample as the reference. (**Page 27, line 650-652**)

Table S3. List of functional groups in SOA extracts. (**Page 4, line 70, Supporting Information**)

| Functional groups | Wavenumber (cm$^{-1}$) |
|---|---|
| C-OH in alcohol | 3360 |
| O-H stretching vibration of carboxylic acid | 3192 |
| C-H stretching vibration of alkane | 2921, 2850 |
| C=O stretching vibrations | 1660, 1633 |
| deformation vibrations of methyl and methylene groups | 1465, 1415 |
| nitrate groups in nitrate ester | 1268 |

4.  *Based on figure 5, you cannot say (line 243) that the mixed AVOCS forms more carboxyl compounds. Because the mixed experiments contain more aerosol mass, you would expect the absorbance to be greater even with the same yield of carboxyls (or nitrate, see comment 3) relative to the non-mixed experiments.*

    The description in the manuscript has been modified, and the mass absorption efficiency (MAE) is applied to better present the light absorption properties of SOA extracts.

    **Page 8-9, line 251-258**: "Based on the light absorption spectra, the mass absorption efficiency (MAE, m$^2$/g) of the SOA in the extracts is calculated using the following equation (Chen et al., 2016):

$$MAE(\lambda) = ln(10)\,Abs\,(\lambda)/C_{OM} \qquad (7)$$

where $Abs(\lambda)$ is the light absorption coefficient ($m^{-1}$), and $C_{OM}$ is the SOA mass concentrations in the extracts. The MAE of the SOA extracts in Figure 6 (a) was calculated from 200 to 300 nm. The MAE at 205 nm were in the order: 1,3,5-TMB SOA (56.8 $m^2/g$) > dodecane SOA (42.5 $m^2/g$) > mixture AVOCs SOA (19.5 $m^2/g$). The MAE in the 210-250 nm band also show the same pattern. This indicates that the SOA generated by the mixture AVOCs contains less light-absorbing substance per unit mass relative to dodecane SOA and 1,3,5-TMB SOA."

5.  *The authors note a handful of peaks from ESI that only appear in mixed experiments, but in Figure 4 it is not especially clear that there are many peaks in panel C that are not in panel A and B. Does the ESI have similar sensitivity for all observed species? Or can sensitivity vary greatly between species? Please address this, it would help the reader to interpret figure 4.*

Although the sensitivity of ESI-ToF-MS to different species may be different, our purpose is not to quantify their intensity or contribution, but to analyze their differences between experiments. Therefore, we believe that the difference in sensitivity does not affect the results and conclusion in this study.

In order to clearly see the magnitude of the peaks unique to the mixed experiments, Figure 4 has been updated as follows:

[Figure]

Figure 4. Mass spectra difference in (a) mixed AVOCs SOA minus *n*-dodecane SOA, and (b) mixed

AVOCs SOA minus 1,3,5-TMB SOA. The Y-axis is the relative intensity normalized by dividing by the total signal strength of the mass spectra. (**Page 25, line 642**)

6. *Also, as it is difficult to see the magnitude of the peaks unique to the mixed experiments in panel C - is it possible that the unique peaks are present in unmixed experiments but just under detectable limits, and upon producing greater aerosol in the mixed experiments, the unique peaks were elevated to above detectable limits? Perhaps a useful way to address this would be give the intensities of the unique peak relative to the larges peak of the spectrum, in addition to addressing comment 5.*

In order to clearly see the magnitude of the peaks, the relative intensity of the observed species have been added in Table 2.

Table 2. Representative identified mass spectral peaks, molecular formulas, molecular weights, and relative intensity of *n*-dodecane, 1,3,5-TMB, and mixture AVOCs-derived SOA. (**Page 21, line 615**)

| Molecular Formula | M+H | M+Na | MIX-Relative Intensity ($\times 10^{-3}$) | TMB-Relative Intensity ($\times 10^{-3}$) | n-dodecane Relative Intensity ($\times 10^{-3}$) |
|---|---|---|---|---|---|
| $C_9H_{14}O_3$ | 171.099 | | 1.41 | 1.73 | 0.733 |
| $C_9H_{17}NO_3$ | | 210.111 | 0.484 | 2.38 | 0.864 |
| $C_{11}H_{18}O_4$ | 215.126 | | 0.641 | 0.0563 | 0.0489 |
| $C_{14}H_{22}O_3$ | 239.166 | | 0.217 | 0.0526 | 0.0489 |
| $C_{14}H_{20}O_2$ | | 243.134 | 0.113 | 0.0966 | 0.110 |
| $C_{11}H_{22}NO_5$ | 249.158 | | 0.489 | 0.0583 | 0.0718 |
| $C_{14}H_{26}O_2$ | | 249.183 | 0.503 | 0.0495 | 0.0459 |
| $C_{13}H_{25}NO_2$ | | 250.177 | 1.22 | 1.79 | 1.69 |
| $C_8H_{12}O_9$ | 253.056 | | 0.220 | 0.0308 | 0.0109 |
| $C_{11}H_{22}O_5$ | | 257.135 | 0.535 | 0.0265 | 0.0223 |
| $C_{12}H_{20}O_6$ | 261.131 | | 0.387 | 0.284 | 0.393 |
| $C_{15}H_{28}O_2$ | | 263.199 | 0.250 | 0.0808 | 0.0427 |
| $C_{14}H_{24}O_5$ | 273.167 | | 0.799 | 0.0872 | 0.0643 |
| $C_{16}H_{22}O_4$ | 279.159 | | 0.728 | 0.139 | 0.0724 |
| $C14H_{26}O_4$ | | 281.172 | 0.305 | 0.0685 | 0.0427 |
| $C_{18}H_{28}O$ | | 283.207 | 1.14 | 1.83 | 1.46 |
| $C_{13}H_{22}NO_6$ | 289.153 | | 0.890 | 0.0513 | 0.0565 |
| $C_{16}H_{22}O_4$ | | 301.146 | 4.77 | 1.09 | 1.41 |
| $C_{18}H_{34}O_2$ | | 305.263 | 3.26 | 1.22 | 0.004.4 |
| $C_{16}H_{30}O_4$ | | 309.202 | 1.28 | 0.149 | 0.135 |
| $C_{18}H_{28}O_3$ | | 315.194 | 0.936 | 1.25 | 1.26 |

| | | | | |
|---|---|---|---|---|
| $C_{16}H_{24}O_5$ | | 319.151 | 2.09 | 0.0290 | 0.0197 |
| $C_{20}H_{34}O_2$ | | 329.246 | 0.319 | 0.0927 | 0.133 |
| $C_{19}H_{38}O_4$ | | 353.267 | 1.42 | 1.46 | 2.42 |
| $C_{24}H_{38}O_4$ | | 413.266 | 2.69 | 1.39 | 1.74 |
| $C_{20}H_{34}O_8$ | | 425.214 | 0.297 | 0.395 | 0.0333 |
| $C_{24}H_{36}NaO_5$ | | 427.245 | 0.107 | 0.108 | 0.0393 |
| $C_{27}H_{48}O_8$ | | 523.325 | 0.183 | 1.54 | 1.33 |
| $C_{30}H_{60}NO_6$ | | 553.459 | 3.08 | 0.145 | 3.32 |
| $C_{28}H_{48}O_{10}$ | | 567.307 | 0.272 | 0.0195 | 0.0215 |
| $C_{29}H_{48}O_{10}$ | | 579.296 | 1.54 | 0.0150 | 0.0167 |
| $C_{35}H_{68}O_{10}$ | | 639.480 | 1.2 | 0.109 | 0.134 |
| $C_{41}H_{60}NO_6$ | | 685.434 | 2.01 | 0.209 | 2.11 |

*Technical corrections:*

7.  *Line 84: Add "The" before "OH precursor"*

    We have added this in the manuscript.

    **Page 3, line 85**: "OH precursor" → "The OH precursor"

8.  *Line 84 and 86: Correct subscripts in molecular formulas.*

    We have corrected the subscripts in the molecular formulas.

    **Page 3, line 86**: "NaNO2" → "NaNO$_2$"

    **Page 3, line 88**: "NO2" → "NO$_2$"

9.  *Line219: m/z is commonly italicized, m/z, this should be corrected throughout document.*

    We have corrected this throughout this manuscript.

    **Page 7, line 218-219; Figure 4; Figure S2**: "m/z" → "*m/z*"

10. *Line 243: "formes" should be forms*

    We have corrected this in the manuscript.

---

## Author Response (AR2)

Dear Professor Roya Bahreini,

Thank you very much for handling our manuscript submitted to *Atmospheric Chemistry and Physics* (**MS No.:** acp-2020-1189; **Title:** Enhanced secondary organic aerosol formation from the photooxidation of mixed anthropogenic volatile organic compounds).

We have addressed all your comments and revised our manuscript very carefully. To proceed, we have uploaded three files, including 1) our point-to-point reply; 2) the revised manuscript with changes highlighted in yellow; 3) the revised manuscript without track-changes.

On behalf of all the co-authors, I would like to thank you and referees for all the invaluable comments. Please feel free to contact me if you need any further information.

Sincerely,

Hong Li, PhD

Chinese Research Academy of Environmental Sciences

Email: lihong@craes.org.cn

1. Wall-loss correction: Particle wall loss is size dependent; however, here the correction is based on integrated mass. If you apply size dependent particle loss correction, how different would the results be?

Thank you very much for this question. The wall loss experiments of inert particles (ammonium sulfate) were performed with the chamber applied in this work, and the size-dependent coefficients were obtained (Li et al., 2021):

$$k_{dep}(d) = 6.35 \times 10^{-6} d^{1.56} + \frac{6.38}{d^{0.67}}$$

With the obtained size-dependent coefficients, we re-calibrated the SOA yields. It is shown that the wall-loss correction based on integrated mass is higher than that based on size-dependent coefficients. As size-dependent particle wall-loss correction is more commonly used (Takekawa et al., 2003; Ng et al., 2007; Chen et al., 2019; Chen et al., 2021), we have changed the correction method in this work, and the corresponding parts in the manuscript have been updated:

(**Page 5, line 136-140**) "The particle growth data was corrected for wall-loss, in which size-dependent coefficients from inert particle wall-loss experiments (ammonium sulfate) were applied to the particle volume data (Li et al., 2021):

$$k_{dep}(d) = 6.35 \times 10^{-6} d^{1.56} + \frac{6.38}{d^{0.67}} \qquad (4)$$

where $k_{dep}(d)$ was the wall-loss loss coefficient of particles in the diameter $d$."

(**Page 21, line 640-647**) "Table 1. Summary of the initial conditions and results of the conducted experiments."

**Table 1. Summary of the initial conditions and results of the conducted experiments.**

| Number | Date | Initial Conditions of the Experiments | | | | | | | | | General Results of the Experiments | | | | | |
|---|---|---|---|---|---|---|---|---|---|---|---|---|---|---|---|---|
| | | 1,3,5-TMB (ppb) | $n$-dodecane (ppb) | NO (ppb) | $NO_2$ (ppb) | $NO_X$ (ppb) | ΔVOCs/NOx (ppbC/ppb) | T (at noon) (°C) | $J(NO_2)$ (at noon) ($s^{-1}$) | RH (%) | $O_3$ (ppb) | Mass[a] ($\mu g/m^3$) | Mass Predicted[b] ($\mu g/m^3$) | Mass Corr.[c] ($\mu g/m^3$) | Mass Predicted[d] ($\mu g/m^3$) | Yield[e] (%) |
| Dod-1 | 2019.09.27 | -- | 22 | 50 | 160 | 210 | 1.26 | 37 | 0.0050 | < 2 | 57 | 6.4 | -- | 35.2 | -- | 23.1 |
| Dod-2 | 2019.10.09 | -- | 20 | 77 | 137 | 214 | 1.12 | 34 | 0.0048 | < 2 | 25 | 3.7 | -- | 19.1 | -- | 14.1 |
| TMB-1 (Li et al., 2021) | 2019.09.03 | 105 | -- | 23 | 188 | 211 | 4.48 | 43 | 0.0056 | < 4 | 288 | 2.1 | -- | 7.68 | -- | 1.5 |
| TMB-2 (Li et al., 2021) | 2019.09.25 | 178 | -- | 46 | 151 | 197 | 8.13 | 38 | 0.0053 | < 4 | 772 | 5.1 | -- | 21.11 | -- | 2.4 |
| TMB-3 (Li et al., 2021) | 2019.10.14 | 170 | -- | 68 | 182 | 250 | 6.12 | 30 | 0.0055 | < 5 | 530 | 2.5 | -- | 8.99 | -- | 1.1 |
| MIX-1 | 2019.09.07 | 168 | 28 | 62 | 169 | 231 | 8 | 46 | 0.0058 | < 3 | 358 | 59.3 | 10.3 | 73.5 | 49.9 | -- |
| MIX-2 | 2019.09.21 | 155 | 22 | 58 | 154 | 212 | 7.83 | 39 | 0.0056 | < 2 | 721 | 47.4 | 8.5 | 58.5 | 40.9 | -- |
| MIX-3 | 2019.09.19 | 182 | 20 | 71 | 147 | 218 | 8.61 | 31 | 0.0044 | < 9 | 435 | 11.5 | 8.6 | 14.8 | 40.8 | -- |
| MIX-4 | 2020.08.21 | 251 | 35 | 54 | 158 | 212 | 12.64 | 39 | | < 7 | 999 | 60.2 | 13.7 | 74.6 | 65.8 | -- |
| MIX-5 | 2020.07.14 | 4 h add | 27 | 61 | 157 | 218 | -- | 52 | 0.0051 | < 5 | 289 | 8 | -- | 27.7 | -- | -- |
| MIX-6 | 2020.07.20 | 4 h add | 38 | 58 | 198 | 256 | -- | 43 | 0.0057 | < 4 | 276 | 6 | -- | 20.8 | -- | -- |
| MIX-7 | 2020.07.24 | 1 h add | 39 | 56 | 207 | 263 | -- | 42 | 0.0058 | < 6 | 440 | 2.3 | -- | 7.9 | -- | -- |
| MIX-8 | 2020.07.22 | 227 | 1h add | 48 | 167 | 216 | -- | 43 | 0.0051 | < 5 | 335 | 4.5 | -- | 15.6 | -- | -- |

a: the mass here is the measured value with the SMPS; the density of the formed SOA derived from 1,3,5-TMB is assumed to be 1.4 $g/cm^3$ (Zhang et al., 2016; Nakao et al., 2013); the density of the formed SOA derived from $n$-dodecane is assumed to be 1.06 $g/cm^3$ (Lim and Ziemann, 2009; Li et al., 2017a); the density of the formed SOA derived from the mixed AVOCs is assumed to be 1.23 $g/cm^3$.

b: the predicted mass here is based on the yield that the particle and vapor wall-loss are not considered.

c: the corrected mass here is calculated after taking particle and vapor wall loss into account.

d: the predicted mass here is based on the yield that the particle and vapor wall-loss are considered.

e: the SOA yield here is calculated after taking particle and vapor wall loss into account.

2. What is the estimated OH in the pure and mixture experiments? How similar is the reactivity with respect to OH during pure and mixture experiments? How does this reactivity compare to that of $NO_2$? Why was so much higher initial conc. of 1,3,5-TMB used compared to dodecane?

Ng et al. (2007) reported that the efficient photolysis of HONO (the same method with this study) could generate relatively high concentrations of OH, 1 ppm $NO_x \sim 2 \times 10^7$ molecules/$cm^3$ OH initially. The $NO_x$ concentration applied in this work is in the range of 190 ~260 ppb, resulting in the estimated OH concentration being $(4 - 5.2) \times 10^6$ molecules/$cm^3$ in the pure and mixture experiments. This part has been added in the manuscript (**Page 6, line 157-161**).

Rate constants for the reactions of n-dodecane and 1,3,5-TMB with OH radical at 298 K are $13.2 \times 10^{-12}$ $cm^3$ molecule$^{-1}$ s$^{-1}$ and $56.7 \times 10^{-12}$ $cm^3$ molecule$^{-1}$ s$^{-1}$, respectively (Atkinson and Arey, 2003). As shown in Table 2, OH reactivity of Dod-1 and Dod-2 is about 6.5-7.1 s$^{-1}$; OH reactivity of TMB-2, and TMB-3 is in the range of 237.1-248.3 s$^{-1}$; and OH reactivity of MIX-1, MIX-2, MIX-3, and MIX-4 is in the range of 223.3-361.5 s$^{-1}$. This indicates that OH reactivity of the mixture experiments differs greatly from that of dodecane experiments, but it is very close to that of 1,3,5-TMB experiments. However, the mixture experiments still have a large enhancement in SOA formation compared with 1,3,5-TMB experiments, indicating that this enhancement is likely not due to the different OH reactivity.

Rate constants for the reactions of $NO_2$ and NO with OH radical at 298 K are $4.1 \times 10^{-11}$ $cm^3$ molecule$^{-1}$ s$^{-1}$ and $3.3 \times 10^{-11}$ $cm^3$ molecule$^{-1}$ s$^{-1}$, respectively (Atkinson et al., 2004). The OH reactivity of $NO_x$ is similar for all experiments (189.6~238.8 s$^{-1}$), and therefore likely plays a minor role in influencing SOA concentration.

And this part has been added in the manuscript (**Page 7, line 213-222).**

**Table 2. OH reactivity (OHR) of the pure and mixture experiments.** (This table has been added in the Supporting Information as Table S3: **Page 4, line 67-70**)

| | 1,3,5-TMB (ppb) | n-dodecane (ppb) | $NO_X$ (ppb) | $OHR_{VOCs}{}^a$ ($s^{-1}$) | $OHR_{NOx}{}^b$ ($s^{-1}$) |
|---|---|---|---|---|---|
| Dod-1 | -- | 22 | 210 | 7.1 | 201.9 |
| Dod-2 | -- | 20 | 214 | 6.5 | 200.7 |
| TMB-1 | 105 | -- | 211 | 146.5 | 208.3 |
| TMB-2 | 178 | -- | 197 | 248.3 | 189.6 |
| TMB-3 | 170 | -- | 250 | 237.1 | 238.8 |
| MIX-1 | 168 | 28 | 231 | 243.4 | 220.8 |
| MIX-2 | 155 | 22 | 212 | 223.3 | 202.4 |
| MIX-3 | 182 | 20 | 218 | 260.4 | 205.9 |
| MIX-4 | 251 | 35 | 212 | 361.5 | 203.2 |

a: $k_{OHR_{VOCs}} = \sum k_{OH+VOC_i}[VOC_i]$

b: $k_{OHR_{NOx}} = k_{OH+NO_2}[NO_2] + k_{OH+NO}[NO]$

The initial concentration ratio of 1,3,5-TMB and n-dodecane in this work is about 10:1 (ppbv), which is mainly based on the following literature research:

Schauer et al. (2002) reported that 1,3,5-TMB and n-dodecane in the gasoline composition were about 7450 and 136 μg/g, respectively; Gentner et al. (2012) reported that the weight percentage of 1,3,5-TMB and n-dodecane in liquid gasoline were 0.530-0.881 and 0.004-0.045 (% weight by carbon), respectively. According to field observations in China, the measured 1,3,5-TMB concentration at the rural site in the YelRD (Yellow River Delta) region in 2017 could reach 1.447 ppb (Chen et al., 2020), and the measured C12 alkane concentration was $0.122 \pm 0.12$ ppb at PRD (Pearl River Delta) region, and 0.129±0.086 ppb at NCP (North China Plain) region in 2018 (Wang et al., 2020).

The concentration of 1,3,5-TMB is much higher than that of n-dodecane in both the gasoline compositions and ambient air. Thus, the initial concentration of 1,3,5-TMB used in this work is much higher than n-dodecane.

This part has been added in the manuscript (**Page 3, line 67-75**).

3. Since the experiments were carried out under high NOx, it's likely that some nitric acid also partitioned in the aerosols. It seems the total volume increase is assigned

to be due to SOA formation. What is the magnitude of the error associated with this assumption?

According to a previous study (Chen et al., 2019), the formed inorganic nitrate is negligible for the high-NOx oxidation of gasoline, in which the experimental conditions are similar to this study (NOx 130 ppb, formed aerosol mass concentration 34.6 $\mu g/m^3$).

In the pure and mixture experiments, the NOx concentration is equivalent, so the formed nitric acid should be similar. Therefore, the increase in particle mass concentration in the mixture experiments is likely from the organic aerosols.

Based on the analysis above, we believe that the error associated with this assumption is small.

This part has been added in the manuscript (**Page 6, line 164-168**).

**Editorial corrections:**

1. L170: sentence after "however" needs to be rephrased.

(**Page 7, line 193**): "however, they only change little on the SOA yield." → "however, the effect is not obvious"

2. L184: change "likely not this case here" to "likely not the case here"

(**Page 7, line 206-207**): "likely not this case here" → "likely not the case here"

3. L194: products "have"

(**Page 8, line 227**): "products has" → "products have"

4. L195: "coagulate" is not the right phrase here; consider "condense" instead

(**Page 8, line 228**): "the products of 1,3,5-TMB to coagulate" → "the products of 1,3,5-TMB to condense"

5. L198: consider changing to "different injection experiments…."

(**Page 8, line 231**): "the injection experiments are performed" → "different injection experiments are performed"

6. L236: change "has" to "have"

This has been corrected in the manuscript.

7. L247: change "are existed" to "exist"

This has been corrected in the manuscript.

8. L248: what do you mean by "and the relative strength is mixed"? Please clarify

The meaning of this sentence is that "and order of the relative strength is mixture AVOCs SOA > n-dodecane SOA > 1,3,5-TMB SOA.". According to the comments of the reviewers, we have updated the content of this part in the latest version.

9. L253-254: consider deleting "In this work" and "mainly"

(**Page 10, line 296**): "In this work, the factors affecting ozone generation considered in this work mainly included……" → "The factors affecting ozone generation included……"

10. L268: the lines before discuss ozone formation; however the sentence starting with "In conclusion", discusses results related to particle formation. This logic doesn't

make sense. Please make sure the sentences are not misplaced with that of the next section, i.e., L280-282 that discuss ozone formation.

Thank you for this comment, the sentence in the manuscript has been corrected.

[revised manuscript text omitted]